# Sports-Traj: A Unified Trajectory Generation Model for Multi-Agent Movement in Sports

**Yi Xu**[1]    **Yun Fu**[1,2]
[1]Department of Electrical and Computer Engineering, Northeastern University
[2]Khoury College of Computer Science, Northeastern University

## Abstract

Understanding multi-agent movement is critical across various fields. The conventional approaches typically focus on separate tasks such as trajectory prediction, imputation, or spatial-temporal recovery. Considering the unique formulation and constraint of each task, most existing methods are tailored for only one, limiting the ability to handle multiple tasks simultaneously, which is a common requirement in real-world scenarios. Another limitation is that widely used public datasets mainly focus on pedestrian movements with casual, loosely connected patterns, where interactions between individuals are not always present, especially at a long distance, making them less representative of more structured environments. To overcome these limitations, we propose a Unified Trajectory Generation model, UniTraj, that processes arbitrary trajectories as masked inputs, adaptable to diverse scenarios in the domain of sports games. Specifically, we introduce a Ghost Spatial Masking (GSM) module, embedded within a Transformer encoder, for spatial feature extraction. We further extend recent State Space Models (SSMs), known as the Mamba model, into a Bidirectional Temporal Mamba (BTM) to better capture temporal dependencies. Additionally, we incorporate a Bidirectional Temporal Scaled (BTS) module to thoroughly scan trajectories while preserving temporal missing relationships. Furthermore, we curate and benchmark three practical sports datasets, ***Basketball-U***, ***Football-U***, and ***Soccer-U***, for evaluation. Extensive experiments demonstrate the superior performance of our model. We hope that our work can advance the understanding of human movement in real-world applications, particularly in sports. Our datasets, code, and model weights are available here.

## 1 Introduction

Understanding multi-agent movement patterns is invaluable across various domains, including autonomous driving (Codevilla et al., 2019; Hazard et al., 2022; Xu et al., 2024; Yang et al., 2024), video surveillance (Cristani et al., 2013; Coşar et al., 2016), and sports analytics (Tuyls et al., 2021; Wang et al., 2024b). To decipher agent movement, these applications rely on tasks such as multi-object tracking (Luo et al., 2021), person re-identification (Ye et al., 2021), trajectory modeling (Nagin, 2010), and action recognition (Zhang et al., 2022; Chi et al., 2023). Among these tasks, trajectory modeling is particularly straightforward and effective for understanding the movements. Despite its inherent challenges, such as the complexity of dynamic environments and the subtle agent interactions, significant advancements have been made recently. These advancements are concentrated in three main areas: trajectory prediction, imputation, and spatial-temporal recovery.

Significant developments have been made recently in modeling trajectories, yet most approaches are specialized for a single task. For example, numerous studies (Xu et al., 2020; 2021; 2022b; Rowe et al., 2023; Mao et al., 2023; Jiang et al., 2023; Chen et al., 2023a; Zhou et al., 2023; Gu et al., 2023; Aydemir et al., 2023; Bae et al., 2023; Shi et al., 2023; Chen et al., 2023b; Seff et al., 2023; Park et al., 2023; Xu et al., 2023a; Maeda & Ukita, 2023; Park et al., 2024; Xu & Fu, 2024) have focused on pedestrian trajectory prediction, driven by the growing interest in autonomous driving, achieving promising results on public datasets. However, these approaches often struggle to generalize to other trajectory-related tasks, such as trajectory imputation and spatial-temporal recovery. These tasks require exploiting both forward and backward spatial-temporal dependencies, which are not typically

addressed in prediction models. Moreover, early datasets like ETH-UCY (Pellegrini et al., 2009; Lerner et al., 2007) and SDD (Robicquet et al., 2016) primarily focus on pedestrian trajectories in real-world scenarios, such as university campuses and sidewalks. In these settings, pedestrians typically move casually, with limited sparse interactions between individuals, particularly those at longer distances. The most common social interactions observed are group walking or collision avoidance. While some studies (Liu et al., 2019; Chib & Singh, 2025) have tackled the multi-agent trajectory imputation problem, they often overlook the future trajectories of agents. This omission limits

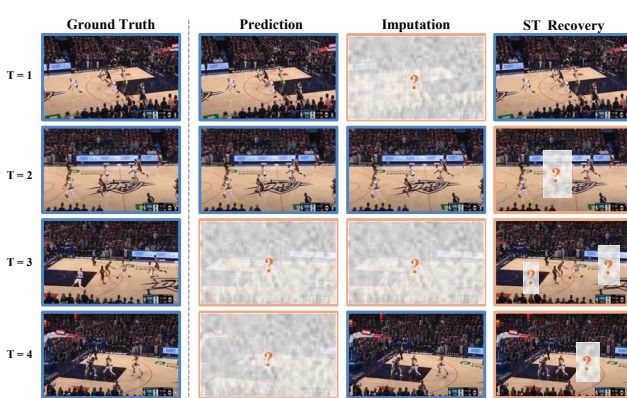

Figure 1: Demonstration of three trajectory modeling tasks, trajectory prediction, imputation, and spatial-temporal (ST) recovery, for multi-agent movement analysis during an offensive possession in a basketball game, where each task takes different inputs.

their practical utility in complete movement understanding, where predicting future trajectories is crucial for downstream planning rather than merely reconstructing historical trajectories. Although some recent efforts (Xu et al., 2023b; Qin et al., 2023) have attempted to integrate imputation and prediction, they both rely on a multi-task framework, and the masking strategies fall short in handling diverse missing patterns. Given that any situation might happen in real practice, it is crucial to develop a general method that can accommodate various scenarios as shown in Figure 1. This raises two pivotal questions: **(1)** *How can we unify these disparate but relevant tasks into a general framework that accommodates different settings?* **(2)** *How can we effectively model these trajectories with varying input formulations?*

Prompted by these questions, we revisit these tasks and introduce the **Uni**fied **Traj**ectory Generation model (UniTraj), which integrates these tasks into a general scheme. Specifically, we merge different input types into a single unified formulation: an arbitrary incomplete trajectory with a mask that indicates the visibility of each agent at each time step. We uniformly process the inputs of each task as masked trajectories, aiming to generate complete trajectories based on the incomplete ones. To model spatial-temporal dependencies across various trajectory formulations, we introduce a Ghost Spatial Masking (GSM) module embedded within a Transformer-based encoder for spatial feature extraction. Leveraging the notable capability of recent popular State Space Models (SSMs), namely the Mamba model, we adapt and enhance it into a Bidirectional Temporal Mamba encoder for long-term multi-agent trajectory generation. Furthermore, we propose a simple yet effective Bidirectional Temporal Scaled (BTS) module that comprehensively scans trajectories while preserving the integrity of temporal relationships within the sequence. Due to the lack of densely structured trajectory datasets, we benchmark and release three sports datasets: ***Basketball-U***, ***Football-U***, and ***Soccer-U***, to facilitate evaluation. In summary, the contributions of our work are as follows:

- We propose UniTraj, a unified trajectory generation model capable of addressing diverse trajectory-related tasks such as trajectory prediction, imputation, and spatial-temporal recovery while handling various input formulations, setting constraints, and task requirements.

- We introduce a novel Ghost Spatial Masking (GSM) module and extend the Mamba model with our innovative Bidirectional Temporal Scaled (BTS) module to extract comprehensive spatial-temporal features from different incomplete trajectory inputs.

- We curate and benchmark three sports datasets, ***Basketball-U***, ***Football-U***, and ***Soccer-U***, and establish strong baselines for this integrated challenge.

- Extensive experiments validate the consistent and exceptional performance of our method.

## 2 PROPOSED METHOD

### 2.1 PROBLEM DEFINITION

To handle various input conditions within a single framework, we introduce a unified trajectory generative model that treats any arbitrary input as a masked trajectory sequence. The trajectory's visible regions are used as constraints or input conditions, whereas the missing regions are the targets for our generative task. We provide the following problem definitions.

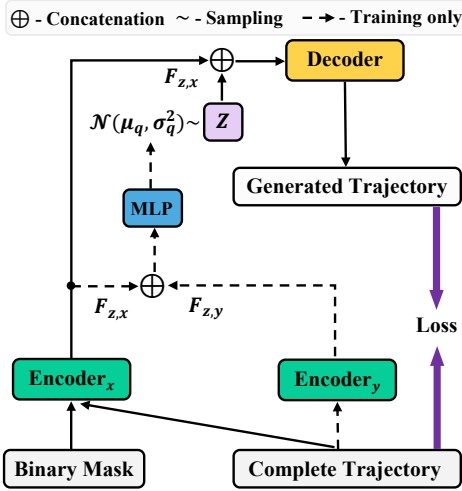

Consider a complete trajectory $X \in \mathbb{R}^{N \times T \times D}$, where $N$ is the number of agents, $T$ represents the trajectory length, and $D$ is the dimension of the agents' states. We denote the position of agent $i$ at time $t$ as $\boldsymbol{x}_i^t \in \mathbb{R}^D$. Typically, we set $D = 2$, corresponding to the 2D coordinates. We also utilize a binary masking matrix $M \in \mathbb{R}^{N \times T}$, valued in $\{0, 1\}$, to indicate missing locations. The variable $m_i^t$ is set to 1 if the location of agent $i$ is known at time $t$ and 0 otherwise. In our work, if an agent is missing, both coordinates are missing. The trajectory is therefore divided by the mask into two segments: the visible region defined as $X_v = X \odot M$ and the missing region defined as $X_m = X \odot (\mathbf{1} - M)$. Our task aims to generate a complete trajectory $\hat{Y} = \{\hat{X}_v, \hat{X}_m\}$, where $\hat{X}_v$ is the reconstructed trajectory and $\hat{X}_m$ is

Figure 2: Overall architecture of our UniTraj model. The encoders extract agent features and derive latent variables, while the decoder generates the complete trajectory using the sampled latent variables and agent features.

the newly generated trajectory. For consistency, we refer to the original trajectory as the ground truth $Y = X = \{X_v, X_m\}$. More formally, our goal is to train a generative model $f(\cdot)$ with parameters $\theta$ that outputs a complete trajectory $\hat{Y}$. A common approach to estimate model parameter $\theta$ involves factorizing the joint trajectory distribution and maximizing the log-likelihood, as follows:

$$\theta^* = \arg\max_{\theta} \sum_{\boldsymbol{x}^{\leq T} \in \Omega} \log p_{\theta}(\boldsymbol{x}^{\leq T}) = \arg\max_{\theta} \sum_{\boldsymbol{x}^{\leq T} \in \Omega} \sum_{t=1}^{T} \log p_{\theta}(\boldsymbol{x}^t | \boldsymbol{x}^{<t}), \quad (1)$$

where $\Omega = \{1, 2, ..., N\}$ is the set of agents, and $\boldsymbol{x}^{\leq T}$ represents the agent sequential trajectory.

### 2.2 UNIFIED TRAJECTORY GENERATION MODEL

**Overall Architecture.** The overall high-level architecture of our UniTraj is shown in Figure 2, which illustrates the data flow during the training and testing phases. More specifically, the detailed architecture of the encoder is presented in Figure 3. This encoder incorporates a Transformer encoder equipped with a Ghost Spatial Masking (GSM) module and a Mamba encoder enhanced by a Bidirectional Temporal Scaled (BTS) module.

We employ the Conditional Variational Autoencoder (CVAE) framework to model the stochastic behavior of each agent. To learn the model parameters $\theta$, we train our model by maximizing the sequential evidence lower-bound (ELBO), defined as follows:

$$\mathbb{E}_{q_{\phi}(\boldsymbol{z}^{\leq T} | \boldsymbol{x}^{\leq T})} \left[ \sum_{t=1}^{T} \log p_{\theta}(\boldsymbol{x}^t | \boldsymbol{z}^{\leq t}, \boldsymbol{x}^{<t}) - \text{KL}\left(q_{\phi}(\boldsymbol{z}^t | \boldsymbol{x}^{\leq t}, \boldsymbol{z}^{<t}) \| p_{\theta}(\boldsymbol{z}^t | \boldsymbol{x}^{<t}, \boldsymbol{z}^{<t})\right) \right], \quad (2)$$

where $\boldsymbol{z}$ represents the latent variables for all agents. $p_{\theta}(\boldsymbol{z}^t | \boldsymbol{x}^{<t}, \boldsymbol{z}^{<t})$ is the conditional prior of $\boldsymbol{z}$, which is set as a Gaussian distribution. The encoding process is implemented through $q_{\phi}(\boldsymbol{z}^t | \boldsymbol{x}^{\leq t}, \boldsymbol{z}^{<t})$, and the decoding process through $p_{\theta}(\boldsymbol{x}^t | \boldsymbol{z}^{\leq t}, \boldsymbol{x}^{<t})$. Note that Equation 2 is a lower bound of the log-likelihood described in Equation 1 and is calculated by summing the CVAE ELBO across each time step.

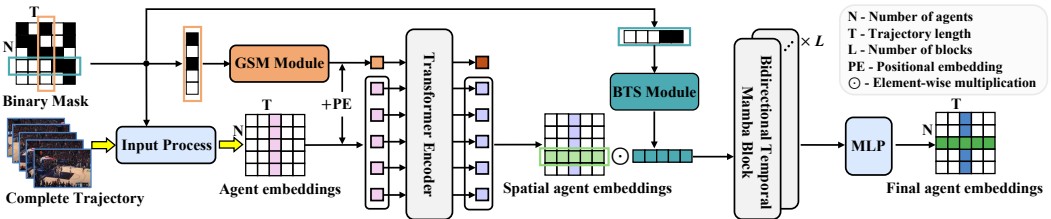

Figure 3: Detailed architecture of the encoding process, which consists of two main components: a Transformer encoder equipped with the GSM model, and a Mamba-based encoder featuring the BTS module. These components are designed to capture comprehensive spatial-temporal features and enable the model to learn missing patterns, thus generalizing to various missing situations.

**Input Process.** Consider an agent $i$ at time step $t$ with position $\boldsymbol{x}_i^t$. We first compute the relative velocity $\boldsymbol{v}_i^t$ by subtracting the coordinates of adjacent time steps. For missing locations, we fill in the values using $(0, 0)$ by element-wise multiplication with the mask. Additionally, we define a one-hot category vector $\boldsymbol{c}_i^t \in \mathbb{R}^3$ to represent three agent categories: ball, offensive player, or defensive player. This categorization is crucial in sports games where players may adopt specific offensive or defensive strategies. The agent features are projected to a high-dimensional feature vector $\boldsymbol{f}_{i,x}^t$. During training, we compute the feature vector $\boldsymbol{f}_{i,y}^t$ for the ground truth trajectory using similar steps but without the multiplication with the mask. Note that $\boldsymbol{f}_{i,y}^t$ is not computed during testing. The input feature vectors are calculated as follows:

$$
\begin{aligned}
\boldsymbol{f}_{i,x}^t &= \varphi_x \left( (\boldsymbol{x}_i^t \odot m_i^t) \oplus (\boldsymbol{v}_i^t \odot m_i^t) \oplus m_i^t \oplus \boldsymbol{c}_i^t; \mathbf{W}_x \right) \\
\boldsymbol{f}_{i,y}^t &= \varphi_y \left( \boldsymbol{x}_i^t \oplus \boldsymbol{v}_i^t \oplus \mathbf{1} \oplus \boldsymbol{c}_i^t; \mathbf{W}_y \right)
\end{aligned}
, \tag{3}
$$

where $\varphi_x(\cdot)$ and $\varphi_y(\cdot)$ are projection functions with weights $\mathbf{W}_x \in \mathbb{R}^{8 \times D}$ and $\mathbf{W}_y \in \mathbb{R}^{8 \times D}$, $\odot$ represents element-wise multiplication, and $\oplus$ indicates concatenation. We implemented $\varphi_x(\cdot)$ and $\varphi_y(\cdot)$ using an MLP. This approach allows us to incorporate location, velocity, visibility, and category information for extracting spatial features for subsequent analyses.

**Transformer Encoder with Ghost Spatial Masking Module.** Unlike other sequential modeling tasks, it is crucial to consider dense social interactions, especially in sports games. Existing studies on human interactions predominantly use attention mechanisms like cross-attention and graph attention to capture these dynamics. However, given that we are addressing a unified problem with arbitrary incomplete inputs, it is essential for our model to learn the spatial-temporal missing patterns. While some studies (Zhao et al., 2022; Xu et al., 2023b) utilize edge graphs with attached labels to extract missing features, this approach is resource-intensive and requires the construction of an additional graph layer. In contrast, we propose a novel and efficient Ghost Spatial Masking (GSM) module to abstract and summarize the spatial structures of missing data. This module can be smoothly integrated into the Transformer without complicating the model structure.

The Transformer (Vaswani et al., 2017) was originally proposed to model the temporal dependencies for sequential data, and we adopt the multi-head self-attention design along the spatial dimension. At each time step, we treat each agent's embedding as a token and input $N$ agent tokens into the Transformer encoder to capture the attentive relationships among the agents' features. In this way, agent-agent interactions are computed through the multi-head self-attention mechanism. However, it is important to note that at each time step, the missing patterns vary, with some players considered absent. A straightforward approach is to encode these missing features at each time step as well. Given the current mask vector $\boldsymbol{m}^t \in \mathbb{R}^{N \times 1}$ of $N$ agents at time step $t$, our goal is to generate a masking embedding based on this mask. We propose a simple yet effective method to generate the so-called ghost masking embedding as follows:

$$
\boldsymbol{f}_{gho}^t = \max_{i=1}^{N} \left( \text{MLP} \left( \text{repeat}(\boldsymbol{m}^t, D)[i, :] \right) \right), \tag{4}
$$

where the repeat$(\cdot)$ function expands the mask vector into a $N \times D$ matrix by repeating it $D$ times, MLP is a linear layer used to project the mask matrix, and max-pooling is applied across rows to derive the ghost masking embedding $\boldsymbol{f}_{gho}^t \in \mathbb{R}^{1 \times D}$. We also explore alternative operations, such as mean or sum pooling, as described in Section 3.3, which also perform effectively in our model.

Once we obtain the ghost masking embedding, we place it at the forefront of the agent embeddings and treat it as an additional head token. This approach is designed to extract order-invariant spatial features of the agents, accommodating any possible arrangement of agent order that might occur in practice. Consequently, we opt to omit the sinusoidal positional embeddings and instead use fully learnable positional embeddings $F_{pos}^t \in \mathbb{R}^{(N+1) \times D}$. The Transformer encoder is defined as follows:

$$
\begin{aligned}
F_{agent}^t &= \boldsymbol{f}_{gho}^t \oplus \boldsymbol{f}_1^t \oplus \boldsymbol{f}_2^t, ..., \oplus \boldsymbol{f}_N^t \\
Q^t, K^t, V^t &= \text{MLP}(F_{agent}^t + F_{pos}^t) \\
F_s^t &= \text{FFN}\left(\text{MultiHeadAttn}(Q^t, K^t, V^t)\right)
\end{aligned}
\quad , \tag{5}
$$

where MultiHeadAttn$(\cdot, \cdot, \cdot)$ represents the multi-head attention layer, and FFN$(\cdot)$ is a feed-forward network consisting of two linear layers and a non-linear activation function. For simplicity, we omit the subscript $x$ or $y$ from the features in Equation 5 since the same model structure is used.

Ultimately, the Transformer encoder outputs the spatial features $F_{s,x}^t$ and $F_{s,y}^t$ for all agents at each time step $t$. We then remove the first token embedding and concatenate these features along the time dimension to obtain the spatial features $F_{s,x} = \{\oplus F_{s,x}^t | t \in 1, 2, ...T\} \in \mathbb{R}^{N \times T \times D}$ and $F_{s,y} = \{\oplus F_{s,y}^t | t \in 1, 2, ...T\} \in \mathbb{R}^{N \times T \times D}$ for the entire trajectory.

**Bidirectional Temporal Mamba with Bidirectional Temporal Scaled Module.** Considering the Mamba model's ability to capture long-term temporal dependencies, we have adapted it to integrate into our framework. However, adapting the Mamba model to our unified trajectory generation task is challenging, primarily due to the lack of an architecture specifically tailored to model trajectories. Effective trajectory modeling requires a thorough capture of spatial-temporal features, which is complicated by the incomplete nature of the trajectories in our task.

To enhance the temporal features extraction while reserving missing relationships, we introduce a Bidirectional Temporal Mamba. This adaptation incorporates multiple residual Mamba blocks paired with an innovative Bidirectional Temporal Scaled (BTS) module. The insight behind this design is that the Mamba blocks are used to extract temporal features, but it is also essential to learn the missing patterns of each trajectory, as these patterns vary across different players. The BTS module is specifically designed to capture these temporal missing patterns, embedding them within the Mamba blocks to ensure more accurate temporal modeling.

Initially, we process the mask $M$ for the entire trajectory by reversing it along the time dimension to produce $\overleftarrow{M}$, which facilitates the model's learning of temporal missing relationships by utilizing both the original and flipped masks within our BTS module. This process generates the scaling matrix $\overrightarrow{S}$ and its corresponding reversed version $\overleftarrow{S}$. Specifically, for agent $i$ at time step $t$, $s_i^t$ is computed as follows:

$$
s_i^t = \begin{cases} 1 + s_i^{t-1} & \text{if } t > 1 \text{ and } m_i^t = 0 \\ 1 & \text{if } t > 1 \text{ and } m_i^t = 1 \\ 0 & \text{if } t = 1 \end{cases} , \tag{6}
$$

where the flip scaling matrix takes the same calculation. Then, we project the scaling matrix $\overrightarrow{S}$ and flipped scaling matrix $\overleftarrow{S}$ to the feature matrix as follows:

$$
\overleftrightarrow{F}_{bts} = 1/\exp\left(\varphi_s(\overleftrightarrow{S}; \mathbf{W}_s)\right), \tag{7}
$$

where $\varphi_s(\cdot)$ is a projection function with weights $\mathbf{W}_s$, $\overleftrightarrow{S} = [\overrightarrow{S}, \overleftarrow{S}]$, and $\overleftrightarrow{F}_{bts} = [\overrightarrow{F}_{bts}, \overleftarrow{F}_{bts}]$. We implement $\varphi_s(\cdot)$ using an MLP with the ReLU activation function.

Equation 6 is designed to calculate the distance from the last observation to the current time step, which helps in quantifying the influence of temporal gaps, particularly when dealing with complex missing patterns. The insight is that the influence of a variable that has been missing for a period decreases over time. Therefore, we utilize a negative exponential function with ReLU to ensure that the influence decays monotonically within a reasonable range between 0 and 1. Accordingly, the transformation function in Mamba model (Gu & Dao, 2023) is revised as follows:

$$
\begin{aligned}
F_{z,x} &= (\overrightarrow{F}_{s,x} \odot \overrightarrow{F}_{bts}) * \overline{\mathbf{K}}_{\mathbf{forw,x}} + \text{Flip}\left((\overleftarrow{F}_{s,x} \odot \overleftarrow{F}_{bts}) * \overline{\mathbf{K}}_{\mathbf{back,x}}\right) \\
F_{z,y} &= \overrightarrow{F}_{s,y} * \overline{\mathbf{K}}_{\mathbf{forw,y}} + \text{Flip}\left(\overleftarrow{F}_{s,y} * \overline{\mathbf{K}}_{\mathbf{back,y}}\right)
\end{aligned}
\quad , \tag{8}
$$

where $\overline{\mathbf{K}}_{\mathbf{forw,x}}$ and $\overline{\mathbf{K}}_{\mathbf{back,x}}$ are convolution kernels in the forward and backward directional Mamba blocks for masked inputs, while $\overline{\mathbf{K}}_{\mathbf{forw,y}}$ and $\overline{\mathbf{K}}_{\mathbf{back,y}}$ are the corresponding convolution kernels for ground truth trajectory features that only calculated in the training phase. The Flip($\cdot$) operation reverses the output features to ensure proper alignment and aggregation.

**Posterior.** The encoding process described above is designed to determine the parameters of the Gaussian distribution for the approximate posterior. Specifically, the mean $\boldsymbol{\mu}_q$ and standard deviation $\boldsymbol{\sigma}_q$ of the posterior Gaussian distribution are calculated as follows:

$$[\boldsymbol{\mu}_q, \boldsymbol{\sigma}_q] = \varphi_q(F_{z,x} \oplus F_{z,y}), \tag{9}$$

where $\varphi_q$ is implemented using an MLP. We sample latent variables $\boldsymbol{Z} \sim \mathcal{N}(\boldsymbol{\mu}_q, \mathrm{Diag}(\boldsymbol{\sigma}_q^2))$ for trajectory generation. During testing, we sample $\boldsymbol{Z}$ from the prior Gaussian distribution $\mathcal{N}(0, \mathbf{I})$.

**Decoder.** To improve the model's ability to generate plausible trajectories, we concatenate the feature $F_{z,x}$ with the latent variable $\boldsymbol{Z}$ before feeding it into the decoder. The trajectory generation process is then calculated as follows:

$$\hat{Y} = \varphi_{dec}(F_{z,x} \oplus \boldsymbol{Z}), \tag{10}$$

where $\varphi_{dec}$ is the decoder function implemented using an MLP.

**Loss Function.** Given an arbitrary incomplete trajectory, our model will generate a complete trajectory. In addition to the ELBO loss defined in Equation 2, we compute the reconstruction loss for the visible regions and include a Winner-Take-All (WTA) loss among a total $K$ generated trajectories to promote generation diversity. These losses are defined as follows:

$$\mathcal{L}_{elbo} = \|\hat{X}_m - X_m\|_2^2 + \lambda_1 \mathrm{KL}\left(\mathcal{N}(\boldsymbol{\mu}_q, \mathrm{Diag}(\boldsymbol{\sigma}_q^2)) \| \mathcal{N}(0, \mathbf{I})\right)$$
$$\mathcal{L}_{rec} = \|\hat{X}_v - X_v\|_2^2 \qquad \mathcal{L}_{wta} = \min_K \|\hat{Y}^{(k)} - Y\|_2^2 \qquad , \tag{11}$$
$$\mathcal{L} = \mathcal{L}_{elbo} + \lambda_2 \mathcal{L}_{rec} + \lambda_3 \mathcal{L}_{wta}$$

where $\lambda_1$, $\lambda_2$, and $\lambda_3$ are hyperparameters used to balance different loss terms. For the WTA loss, we generate multiple trajectories, and $\hat{Y}^{(k)}$ denotes the $k^{\mathrm{th}}$ generated trajectory.

# 3 EXPERIMENTS

## 3.1 BENCHMARKS AND SETUP

**Datasets.** We curate and benchmark three sports game datasets for this integrated challenging trajectory generation task, **Basketball-U**, **Football-U**, and **Soccer-U**. **(1) Basketball-U:** We build Basketball-U from NBA dataset (Zhan et al., 2018), with 93,490 training sequences and 11,543 testing sequences. Each sequence consists of trajectories for 1 ball, 5 offensive players, and 5 defensive players. **(2) Football-U:** Football-U is created from NFL Football Dataset [1], with 10,762 training sequences and 2,624 testing sequences. Each sequence consists of trajectories for 1 ball, 11 offensive players, and 11 defensive players. **(3) Soccer-U:** SoccerTrack [2] (Scott et al., 2022) dataset is used as the base to build Soccer-U. The top-view scenarios are used to extract 9,882 training sequences and 2,448 testing sequences. Each sequence consists of trajectories for 1 ball, 11 offensive players, and 11 defensive players. To cover various input situations, we design different 5 masking strategies. In our datasets, we set the trajectory length to $T = 50$, and the number of agents $N$ includes all the players and the ball. Details can be found in Appendix C.

**Baselines.** We use the following three categories of methods for comparison. Although some of these models were not originally designed for our trajectory generation task, we have adapted them to fit our needs. **(1) Statistical approach:** This includes basic methods such as Mean, Median, and Linear Fit, **(2) Vanilla models:** These are fundamental networks such as LSTM (Hochreiter & Schmidhuber, 1997) and Transformer (Vaswani et al., 2017), **(3) Advanced baselines:** Advanced deep learning models such as MAT (Zhan et al., 2018), Naomi (Liu et al., 2019), INAM (Qi et al., 2020), SSSD (Alcaraz & Strodthoff, 2022), and GC-VRNN (Xu et al., 2023b).

---

[1] https://github.com/nfl-football-ops/Big-Data-Bowl
[2] https://github.com/AtomScott/SportsLabKit

Table 1: We compare our UniTraj with baseline methods and report five metrics on the Basketball-U and Football-U datasets. The best results are highlighted and the second best results are underlined.

| Method | Basketball-U (In Feet) | | | | | Football-U (In Yards) | | | | |
|---|---|---|---|---|---|---|---|---|---|---|
| | minADE$_{20}$ ↓ | OOB ↓ | Step | Path-L | Path-D | minADE$_{20}$ ↓ | OOB ↓ | Step | Path-L | Path-D |
| Mean | 14.58 | **0** | 0.99 | 52.39 | 737.58 | 14.18 | **0** | 0.52 | 25.06 | 606.07 |
| Medium | 14.56 | **0** | 0.98 | 51.80 | 743.36 | 14.23 | **0** | 0.52 | 24.96 | 600.22 |
| Linear Fit | 13.54 | 4.47e-03 | 0.56 | 42.86 | 453.38 | 12.66 | 1.49e-04 | **0.17** | **15.83** | 207.57 |
| LSTM (1997) | 7.10 | 9.02e-04 | 0.76 | 58.48 | 449.58 | 7.20 | 2.24e-04 | 0.43 | 34.06 | 228.13 |
| Transformer (2017) | 6.71 | 2.38e-03 | 0.79 | 59.34 | 517.54 | 6.84 | 5.68e-04 | 0.42 | 33.01 | 202.10 |
| MAT (2018) | 6.68 | 1.36e-03 | 0.88 | 58.83 | 483.46 | 6.36 | 4.57e-04 | 0.40 | 31.32 | 186.11 |
| Naomi (2019) | 6.52 | 2.02e-03 | 0.81 | 69.10 | 450.66 | 6.77 | 7.66e-04 | 0.67 | 42.74 | 259.11 |
| INAM (2020) | 6.53 | 3.16e-03 | 0.70 | 58.54 | 439.87 | 5.80 | 8.30e-04 | 0.39 | 32.10 | 177.04 |
| SSSD (2022) | 6.18 | 1.82e-03 | 0.47 | 46.87 | 393.12 | 5.08 | 6.81e-04 | 0.39 | 23.10 | 122.42 |
| GC-VRNN (2023b) | 5.81 | 9.28e-04 | 0.37 | 28.08 | 235.99 | 4.95 | 7.12e-04 | 0.29 | 32.48 | 149.87 |
| Ground Truth | 0 | 0 | 0.17 | 37.61 | 269.49 | 0 | 0 | 0.03 | 12.56 | 76.68 |
| **UniTraj (Ours)** | **4.77** | 6.12e-04 | **0.27** | **34.25** | **240.83** | **3.55** | 1.12e-04 | 0.23 | 19.26 | **114.58** |

**Evaluation Protocol.** We generate a total of $K = 20$ trajectories based on the masked trajectory input and use the minADE$_{20}$ as one of the evaluation metrics. Furthermore, we include four additional metrics (Zhan et al., 2018; Qi et al., 2020) to comprehensively assess the generation quality. **(1) minADE$_{20}$:** Calculate the minimum average displacement error between the generated trajectories and the ground truth across the 20 generated trajectories, **(2) Out-of-Boundary (OOB):** Measure the percentage of generated locations that fall outside the predefined court boundaries, **(3) Step:** Calculate the average change in step size across the generated trajectories, **(4) Path-L:** Calculate the average length of the trajectories for each agent, **(5) Path-D:** Calculate the maximum difference in trajectory lengths among the agents. Note that except for the metrics **minADE$_{20}$** and **OOB**, where lower values are better, other metrics typically perform better the closer they are to the ground truth. Implementation details are provided in Appendix D.

## 3.2 MAIN RESULTS

Evaluation results using five metrics across three datasets are presented in Table 1 for Basketball-U and Football-U, and in Table 2 for Soccer-U. Across all datasets, our UniTraj model achieves the lowest values on the minADE$_{20}$ metric, confirming its ability to generate trajectories that are closest to the ground truth. Specifically, UniTraj outperforms GC-VRNN by $17.9\%$ on the minADE$_{20}$ metric for the Basketball-U dataset, $28.3\%$ for Football-U, and $10.7\%$ for Soccer-U. Regarding the OOB metric, the Mean and Median methods fill missing positions with

Table 2: We compare UniTraj with baseline methods and report five metrics on the Soccer-U dataset. The best results are highlighted and the second best results are underlined.

| Method | Soccer-U (In Pixels) | | | | |
|---|---|---|---|---|---|
| | minADE$_{20}$ ↓ | OOB ↓ | Step | Path-L | Path-D |
| Mean | 417.68 | **0** | 4.32 | 213.05 | 8022.51 |
| Medium | 418.06 | **0** | 4.39 | 214.55 | 8041.98 |
| Linear Fit | 398.34 | **0** | **0.70** | **112.34** | **2047.19** |
| LSTM (1997) | 186.93 | 4.74e-05 | 7.50 | 652.98 | 4542.78 |
| Transformer (2017) | 170.94 | 6.59e-05 | 6.66 | 566.14 | 4269.08 |
| MAT (2018) | 170.46 | 7.56e-05 | 6.45 | 562.44 | 3953.34 |
| Naomi (2019) | 145.20 | 8.78e-05 | 7.47 | 649.62 | 4414.99 |
| INAM (2020) | 134.86 | 4.04e-05 | 6.37 | 547.02 | 4102.37 |
| SSSD (2022) | 118.71 | 4.51e-05 | 5.11 | 425.98 | 3252.66 |
| GC-VRNN (2023b) | 105.87 | 1.29e-05 | 4.92 | 506.32 | 3463.26 |
| Ground Truth | 0 | 0 | 0.52 | 112.92 | 951.00 |
| **UniTraj (Ours)** | **94.59** | 3.31e-06 | 4.52 | 349.73 | 2805.79 |

mean and median values respectively, naturally avoiding generating out-of-bound locations. Except for these two methods, ours can achieve lower rates across all three datasets, indicating its effectiveness in generating trajectories that remain within the court.

An interesting observation is that the Linear Fit method yields results very close to the ground truth for metrics such as Step, Path-L, and Path-D, comparable to our UniTraj and other deep-learning methods. This is because these metrics do not assess the quality of the locations but rather the length information of the trajectory. For instance, the Step metric measures the second-order information of each step size, where the ground truth is very small, making Linear Fit a suitable method. In comparison, our UniTraj consistently outperforms other deep-learning methods by a large margin, proving its effectiveness. These findings verify the robust performance of UniTraj.

Table 3: Ablation study on three datasets.

| Variants | minADE$_{20}$ ↓ | | |
|---|---|---|---|
| | *Basketball-U* | *Football-U* | *Soccer-U* |
| w/o GSM | 4.86 | 3.92 | 119.43 |
| w/o BTS | 4.86 | 3.60 | 105.47 |
| uni w/ BTS | 5.86 | 4.09 | 106.22 |
| uni w/o BTS | 5.86 | 4.10 | 113.49 |
| **whole (ours)** | **4.77** | **3.55** | **94.59** |

Table 4: Results for different model designs.

| Variants | minADE$_{20}$ ↓ | | |
|---|---|---|---|
| | *Basketball-U* | *Football-U* | *Soccer-U* |
| w/ global | 4.86 | 3.74 | 106.77 |
| w/ learnable | 4.92 | 3.63 | 107.50 |
| w/ mean | 4.80 | 3.64 | 100.84 |
| w/ sum | 4.79 | 3.56 | 102.99 |
| **w/ max (ours)** | **4.77** | **3.55** | **94.59** |

## 3.3 ABLATION STUDY

**Ablation of Each Component.** We start by analyzing the contribution of each component in our UniTraj. Table 3 presents the results of the ablation study for each component. The "w/o GSM" variant omits the Ghost Spatial Masking (GSM) module, feeding only the agent embedding into the Transformer. The "w/o BTS" excludes the Bidirectional Temporal Scaled (BTS) module, relying solely on the bidirectional temporal mamba encoder. The "uni w/ BTS" and 'uni w/o BTS" variants use the forward unidirectional mamba encoder, with and without the BTS module.

Comparing our full model with the "w/o GSM" variant, we see that our proposed GSM module enhances the learning of the masking pattern to boost spatial features. The comparison between the "w/o BTS" variant and the "uni w/ BTS" versus "uni w/o BTS" variants demonstrates the effectiveness of the BTS module in capturing temporal missing patterns, regardless of whether the approach is bidirectional or unidirectional. Additionally, removing the backward mamba block leads to a decrease in performance, validating the effectiveness of our bidirectional design in capturing more comprehensive temporal dependencies. An interesting observation is that the contributions of certain components are different across three datasets. A potential reason is the differences in the number of players and the court sizes in these sports, which may cause the importance of spatial and temporal features to shift accordingly.

**Ghost Masking Embedding.** We explore various strategies for generating a ghost masking embedding. Table 4 presents the results of our UniTraj using different head embeddings. In the "w/ global" variant, we repeat the mask and directly element-wise multiply with the agent embedding to create a hyper embedding that captures global feature information. The "w/ learnable" variant involves replacing our ghost embedding with a fully learnable one. The "w/ mean" and "w/ sum" variants apply mean-pooling and sum-pooling in Equation 4, respectively.

Comparison with the first two variants reveals that directly mapping the mask in UniTraj yields lower ADE across all datasets, underscoring the effectiveness of our GSM module. Results from the pooling variants are comparable, with max-pooling showing a slight advantage. This indicates that any permutation-invariant operation is effective within our module.

**Mamba Block Depth.** We explore how different depths of temporal mamba block affect performance. Table 5 presents the results for the Basketball-U dataset. We observe that with $L = 4$, it achieves the best performance in minADE$_{20}$ and Step while maintaining competitive results on other metrics. Results for other

Table 5: Results on Basketball-U with different depths.

| Depth | *Basketball-U* (In Feet) | | | | | |
|---|---|---|---|---|---|---|
| | minADE$_{20}$ ↓ | OOB ↓ | Step | Path-L | Path-D | Params |
| L = 1 | 5.14 | 4.38e-03 | 0.38 | **39.36** | **281.40** | 0.55M |
| L = 2 | 4.93 | **1.68e-04** | 0.30 | 35.09 | 141.12 | 0.96M |
| L = 3 | 4.85 | 2.95e-03 | 0.31 | 35.38 | 155.73 | 1.36M |
| L = 4 | **4.77** | 6.14e-04 | **0.27** | 34.25 | 240.83 | 1.77M |
| L = 5 | 4.81 | 1.02e-03 | 0.29 | 35.23 | 172.05 | 2.18M |
| **GT** | 0 | 0 | 0.17 | 37.61 | 269.49 | – |

datasets are detailed in Appendix E, where similar trends are observed. Considering the balance between the number of parameters and overall performance, we set L to 4 across all three datasets.

### 3.3.1 QUALITATIVE RESULTS

We present one visualization example from the Basketball-U dataset, as shown in Figure 4. The trajectories generated by our method are noticeably closer to the ground truth. For the missing locations indicated by red scatter points, our method provides more accurate predictions. Furthermore,

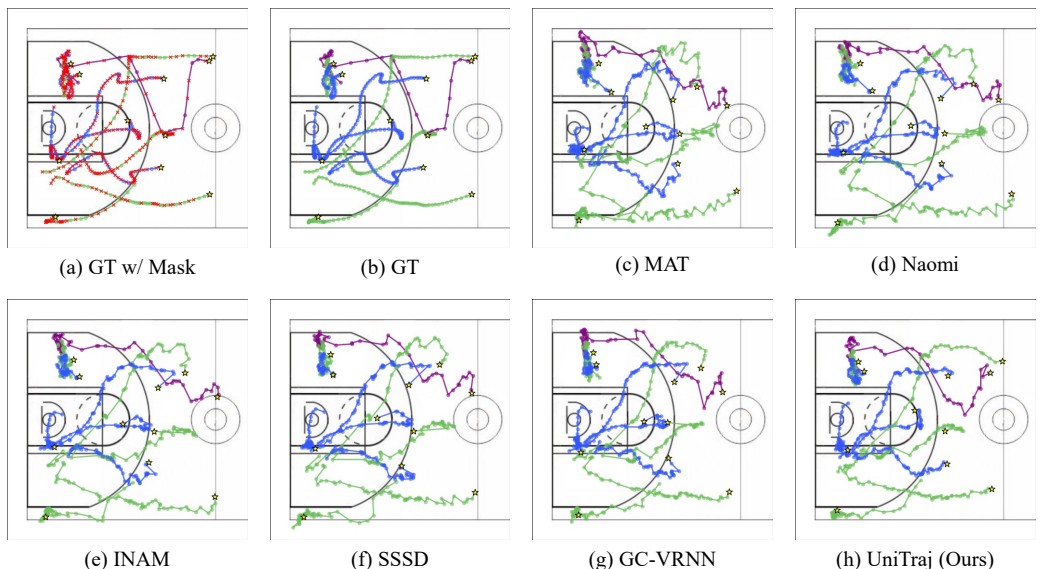

Figure 4: Qualitative comparison between advanced baselines and our method. The ball's trajectory is shown in purple, offensive players are in green, and defensive players are in blue. Red "x" marks indicate masked locations and the starting points of the trajectories are highlighted with yellow stars.

the trajectories produced by our method are significantly smoother compared to the other baselines, which highlights the effectiveness of our proposed method.

## 4    RELATED WORK

**Trajectory Prediction.**    Trajectory prediction aims to predict the future movements of agents conditioned on their historical observations. The biggest challenge is modeling the social interactions among agents, which has driven the development of various methods. A classic approach, Social-LSTM (Alahi et al., 2016), introduced a pooling layer to enable information sharing. Subsequent methods (Vemula et al., 2018; Zhang et al., 2019; Hu et al., 2020; Xu et al., 2021) have employed similar designs to extract comprehensive interaction features. More recent studies (Kosaraju et al., 2019; Sun et al., 2020; Mohamed et al., 2020; Shi et al., 2021; Xu et al., 2022a; Li et al., 2022; Bae et al., 2022) have utilized Graph Neural Networks (GNNs), treating agents as nodes to model social interactions. The inherent uncertainty and diversity of future trajectories have also led to the adoption of generative models, including Generative Adversarial Networks (GANs) (Gupta et al., 2018; Sadeghian et al., 2019; Li et al., 2019; Amirian et al., 2019), Conditional Variational Autoencoders (CVAEs) (Mangalam et al., 2020; Xu et al., 2022b; Ivanovic & Pavone, 2019; Salzmann et al., 2020; Xu et al., 2022c), and Diffusion models (Gu et al., 2022; Mao et al., 2023; Jiang et al., 2023). However, prediction is not always the primary objective in real-world applications such as similarity analysis or action localization. Our work aims to tackle a broader range of challenges within trajectory modeling and introduces a new benchmark to better address these complexities.

**Trajectory Imputation and Spatial-Temporal Recovery.**    Imputation is a classic and extensively explored task, with time-series imputation receiving the most attention. Traditional statistical techniques include replacing missing values with the mean or median value (Acuna & Rodriguez, 2004), as well as methods like linear regression (Ansley & Kohn, 1984), k-nearest neighbors (Troyanskaya et al., 2001; Beretta & Santaniello, 2016), and the expectation-maximization (EM) algorithm (Ghahramani & Jordan, 1993; Nelwamondo et al., 2007). These methods often suffer from limited generalization ability due to their reliance on rigid priors. In response, more flexible frameworks have emerged, employing deep learning techniques for sequential data imputation, such as autoregressive imputation using RNNs (Yoon et al., 2018b; Cao et al., 2018), or the use of GANs, VAEs, and Diffusion models (Yoon et al., 2018a; Luo et al., 2018; 2019; Qi et al., 2020; Miao et al., 2021; Tashiro et al., 2021; Wen et al., 2024; Chen et al., 2024; Yuan & Qiao, 2024) to gen-

erate reconstructed sequences. However, only a few studies have specifically addressed trajectory imputation for multi-agent movements. For instance, AOMI (Liu et al., 2019) introduces a non-autoregressive imputation method, while GMAT (Zhan et al., 2018) develops a hierarchical model to generate macro-intent labels for sequence generation. Additionally, Graph Imputer (Omidshafiei et al., 2021) has utilized forward and backward information to model the distribution of imputed trajectories in soccer games.

The task of trajectory spatial-temporal recovery, while similar to trajectory imputation in its focus on incomplete trajectories, differs in its broader goal of reconstructing complete spatial-temporal sequences. This task requires not only filling gaps but also mining the intrinsic relations between different agents' trajectories, which is crucial for real-world applications like arrival time estimation (Derrow-Pinion et al., 2021; Chen et al., 2022) and trajectory similarity computation (Han et al., 2021). A major benchmark involves recovering trajectory data from GPS points (Chen et al., 2011; Wei et al., 2012; 2024; Chen et al., 2023d), while few studies focus on multi-agent contexts.

Recent research has integrated trajectory imputation and prediction tasks. For instance, INAM (Qi et al., 2020) introduces an imitation learning paradigm that handles both tasks asynchronously, while GC-VRNN (Xu et al., 2023b) proposes a multi-task framework using three different GCN layers and a variational RNN to conduct trajectory imputation and prediction simultaneously. Nevertheless, these studies primarily aim to forecast trajectories based on missing observations, whereas our work aims for a more general goal that is not restricted to specific input formats or tasks. Another closely related work is Traj-MAE (Chen et al., 2023b), which introduces a continual framework to pre-train trajectory and map encoders for vehicle trajectory prediction. However, Traj-MAE focuses primarily on vehicle trajectories, which are more closely tied to maps, whereas our approach targets player trajectories in sports games, where denser social interactions occur.

**State Space Models.** The core concept of state space sequence models (SSMs) (Gu et al., 2021a;b) is to link input and output sequences through a latent state. Recently, a promising variant of SSMs, Mamba architecture (Gu & Dao, 2023), which integrates time-varying parameters, has been introduced to enhance efficiency. This architecture has inspired various recent studies in different computer vision tasks. For instance, in works (Liu et al., 2024; Pei et al., 2024; Wang et al., 2024a; Behrouz et al., 2024; Zhu et al., 2024), Mamba has been applied to process image patches, learn visual representations, and achieve impressive performance in downstream tasks, proving to be an effective backbone. However, the full potential of the Mamba model in trajectory modeling remains underexplored. In this work, we extend the Mamba model to learn temporal dependencies bidirectionally and introduce a Bidirectional Temporal Scaled (BTS) module for a comprehensive scan.

## 5 CONCLUSION

In this work, we focus on the domain of sports and address the problem of modeling multi-agent trajectories by considering various situations in real practice, emphasizing the need for a general approach. To accommodate diverse real-world scenarios, we introduce a unified trajectory generation task that simultaneously handles multiple input situations. Essentially, we treat different inputs, regardless of format, as masked trajectories to generate a complete trajectory. To address various missing data scenarios, we propose a simple yet effective Ghost Spatial Masking module that integrates within the Transformer encoder and a novel Bidirectional Temporal Scaled Module embedded within the extended Bidirectional Temporal Mamba encoder. We have also curated and benchmarked three sports game datasets, ***Basketball-U***, ***Football-U***, and ***Soccer-U***, to evaluate our model and establish robust baselines for future research. Extensive experiments confirm the superior performance of our approach.

**Limitations and Social Impacts** One limitation of our approach is the use of simple MLPs for decoding the trajectory. There is potential for improvement by developing a more powerful decoder. In addition, our sports datasets have a fixed number of agents. How to accommodate datasets with varying numbers of agents remains an area for future exploration. We have not observed any negative societal impacts from UniTraj. Instead, we hope to advance the field by building datasets and establishing strong baselines that encourage further study of the unified trajectory generation task in the sports domain.

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

## A    APPENDIX

## B    DATASETS, CODE, AND MODELS

We have provided the GitHub link in the abstract to our datasets, code, and trained checkpoints. The README.md file includes detailed instructions for downloading the datasets and running the code.

## C    DATASETS DETAILS

Table 6: Number of sequences for five masking strategies applied across all sequences. The average masking rate is also included for three datasets.

| Dataset / Masking Type | | 1 | 2 | 3 | 4 | 5 | Total Num | Masking Rate |
|---|---|---|---|---|---|---|---|---|
| Basketball-U | Train | 18,718 | 18,767 | 18,769 | 18,694 | 18,542 | 93,490 | 52.61% |
| | Test | 2,400 | 2,257 | 2,317 | 2,281 | 2,288 | 11,543 | 52.65% |
| Football-U | Train | 2,202 | 2,195 | 2,086 | 2,093 | 2,186 | 10,762 | 45.99% |
| | Test | 532 | 557 | 511 | 534 | 490 | 2,624 | 46.19% |
| Soccer-U | Train | 1,978 | 2,040 | 1,951 | 1,974 | 1,939 | 9,882 | 46.17% |
| | Test | 487 | 461 | 476 | 498 | 526 | 2,448 | 46.41% |

**Basketball-U.**    The base dataset is available from Stats Perform [3], which is used by recent works (Zhan et al., 2018; Liu et al., 2019; Zhan et al., 2020; Xu et al., 2023b). The original dataset contains 104,003 training sequences and 13,464 testing sequences. Each sequence includes xy-coordinates (in feet) of the ball and 10 players (5 offensive and 5 defensive) for 8 seconds, sampled at a frequency of 6.25 Hz, resulting in sequences of length 50. The coordinates are unnormalized, with (0, 0) positioned at the bottom-left corner of the court. The court measures 94 feet in length and 50 feet in width. We first clean the dataset, removing sequences that fall outside the court boundaries. After cleaning, it consists of 93,490 training and 11,543 testing sequences. We then applied five masking strategies to generate the masks.

**Football-U.**    The base dataset is sourced from Next Gen Stats [4], which includes tracking files from the first six weeks of the 2017 season, and each file records one football game. The dataset includes 91 games, of which 73 are used for training and 18 for testing in our work. Each file provides trajectory sequences with xy-coordinates (in yards). These coordinates are unnormalized, with (0, 0) at the bottom-left corner of the field, which measures 120 yards in length and 53.3 yards in width. After cleaning the dataset, we have 9,882 training sequences and 2,448 testing sequences. Each sequence details the trajectories of 1 ball, 11 offensive players, and 11 defensive players.

**Soccer-U.**    We use the top-view scenarios from SoccerTrack dataset [5] (Scott et al., 2022). The dataset contains 60 tracking files, from which we use 48 for training and 12 for testing. By applying a sliding window of size 4, we extract 9,882 training sequences and 2,448 testing sequences. As the data are provided in pixel coordinates, we use the original coordinates, with the soccer field dimensions set at 3,840 by 2,160 pixels. Each sequence consists of trajectories for 1 ball, 11 offensive players, and 11 defensive players.

**Making Strategies.**    We have designed five masking strategies to cover various input conditions, which are detailed in our "generatedataset.py" script:

1. **"Prediction Mask":** Generates a mask from one point to the end of the sequence for the prediction task, splitting the sequence into observation and prediction parts. We randomly select one of the following time points for splitting each agent's sequence: 25, 30, 35, or 40.

---

[3] https://www.statsperform.com/artificial-intelligence-in-sport/
[4] https://nextgenstats.nfl.com/
[5] https://github.com/AtomScott/SportsLabKit

Table 7: Results on Football-U and Soccer-U with different depths.

| Depth | Football-U (In Yards) | | | | | | Soccer-U (In Pixels) | | | | | |
|---|---|---|---|---|---|---|---|---|---|---|---|---|
| | minADE$_{20}$ ↓ | OOB ↓ | Step | Path-L | Path-D | Params | minADE$_{20}$ ↓ | OOB ↓ | Step | Path-L | Path-D | Params |
| L = 1 | 3.68 | 1.99e-04 | 0.34 | 25.33 | 281.83 | 0.55M | 105.33 | 2.91e-04 | 7.28 | 531.20 | 18784.14 | 0.55M |
| L = 2 | 3.57 | 9.73e-05 | **0.23** | **19.26** | 124.01 | 0.96M | 101.16 | 8.28e-06 | **4.21** | **328.51** | 2100.98 | 0.96M |
| L = 3 | 3.59 | **7.70e-05** | 0.25 | 20.30 | 156.98 | 1.37M | 95.36 | 3.58e-05 | 4.42 | 343.45 | 2091.21 | 1.37M |
| L = 4 | **3.55** | 1.12e-04 | **0.23** | **19.26** | 114.58 | 1.77M | **94.59** | **3.31e-06** | 4.52 | 349.73 | 2805.79 | 1.77M |
| L = 5 | 3.59 | 1.76e-04 | 0.25 | 20.27 | 144.20 | 2.18M | 99.77 | 3.78e-05 | 4.51 | 345.13 | **2070.30** | 2.18M |
| **GT** | 0 | 0 | 0.03 | 12.56 | 76.68 | – | 0 | 0 | 0.52 | 112.92 | 951.00 | – |

2. **"Random Consecutive Mask":** Generates a random consecutive hole for each agent, with the number of holes ranging from 1 to 5 and each hole length randomly set to 3, 4, or 5.

3. **"Random Discrete Mask":** Creates a discrete mask where each location has a 50% to 80% probability of being masked.

4. **"Center Consecutive Mask":** Generates a random consecutive hole centrally placed with a length randomly chosen between 25 to 40.

5. **"Random Agent Mask":** Masks 5 players randomly in each sequence.

These hyperparameters are set to achieve an approximate masking rate of 50%. Adjusting these parameters or combining some of them can further increase the masking rate and the generation difficulty. Each masking strategy has an equal chance of being selected for any given sequence. The statistics for the three datasets are presented in Table 6.

# D IMPLEMENTATION DETAILS AND CONFIGURATIONS

The implementation details for different mapping functions $\varphi(\cdot)$ are provided at their first mention in the main paper. We set $\lambda_1 = \lambda_2 = \lambda_3 = 1$ to balance the losses as described in Equation 11. To ensure reproducibility, we fix the random seed at 2024 across our model during training. We project the input to a dimension of 64 as in Equation 3. In our Transformer encoder, the model dimension is set to 64 and the number of heads to 8. The dimension of the latent variable $Z$ is set to 128. In our Mamba block, the state dimension is 64, the convolution kernel size is 4, the expansion value is 2, and the depth $L$ is 4. The experiments are conducted using PyTorch (Paszke et al., 2019) on an NVIDIA A100 GPU. The model is trained over 100 epochs with a batch size of 128. We use the Adam optimizer (Diederik & Jimmy, 2015) with an initial learning rate of 0.001, which is decayed by 0.9 every 20 epochs.

For the baseline methods Vanilla LSTM and Transformer, we adopt the same input processing as in our UniTraj. The hidden state size for LSTM and the agent embedding dimension are both set to 64. In the Transformer model, the number of heads is 8, which is the same in our UniTraj. For the MAT method (Zhan et al., 2018), we use the official code available at [6]. For Naomi (Liu et al., 2019), the code can be found at [7]. For INAM (Qi et al., 2020), we have tried our best to reproduce their methods as described in their paper, since the authors have not released the code. For SSSD (Alcaraz & Strodthoff, 2022), the code is available at [8]. For GC-VRNN (Xu et al., 2023b), we obtained the code directly from the authors, as mentioned at [9].

# E ADDITIONAL EXPERIMENTS

**Mamba Block Depth.** We provide complete results from our study on the impact of Mamba block depths in the Football-U and Soccer-U datasets. The results are presented in Table 7. A similar trend is observed in these datasets as with Basketball-U. Based on these findings, we have set $L = 4$ across all three datasets in our work.

---

[6] https://github.com/ezhan94/multiagent-programmatic-supervision
[7] https://github.com/felixykliu/NAOMI?tab=readme-ov-file
[8] https://github.com/AI4HealthUOL/SSSD
[9] https://github.com/colorfulfuture/GC-VRNN

**Transformer Depth.** We also study the impact of the number of stacked Transformer layers $l$. Table 8 presents the results on Basketball-U dataset. We observe that a single Transformer layer achieves the best results in minADE$_{20}$, while also having the fewest model parameters. One possible reason is that our Transformer encoder is applied along the agent dimension, which is smaller than the sequence length, making one layer sufficient to extract spatial features effectively. Consequently, we set $l = 1$ for all three datasets to maintain simplicity.

Table 8: Study on depth $l$.

| Depth | Basketball-U (In Feet) | |
|---|---|---|
| | minADE$_{20}$ ↓ | Params |
| $l = 1$ | **4.77** | 1.77M |
| $l = 2$ | **4.77** | 2.06M |
| $l = 3$ | 4.82 | 2.35M |
| $l = 4$ | 4.89 | 2.75M |

**Different Temporal Architecture.** To further validate the effectiveness of the Mamba encoder, we replaced it with LSTM, VRNN, and Transformer models to determine if comparable results could be achieved. The results on the Basketball-U dataset are presented in Table 9. Since these architectures are not specifically designed for missing patterns, we also included a variant of ours without the BTS module.

Table 9: Results with different temporal architecture.

| Variants | Basketball-U (In Feet) |
|---|---|
| | minADE$_{20}$ ↓ |
| w/ LSTM | 5.32 |
| w/ VRNN | 5.29 |
| w/ Transformer | 4.99 |
| w/ Mamba w/o BTS | 4.86 |
| w/ Mamba | **4.77** |

It is observed that the Mamba encoder, even without the BTS module, can achieve the lowest minADE$_{20}$. This proves the effectiveness of the Mamba encoder in modeling temporal dependencies.

**Generalizability.** To further validate the generalizability and robustness of our proposed method, we apply it to other individual trajectory tasks. We conducted experiments on the widely used public datasets ETH-UCY Pellegrini et al. (2009); Lerner et al. (2007) and SDD (Robicquet et al., 2016) for trajectory prediction, comparing our method with recent state-of-the-art baselines: MemoNet (Xu et al., 2022b), FlowChain (Maeda & Ukita, 2023), and EqMotion (Xu et al., 2023a). As shown in Tables 10 and 11, our method outperforms FlowChain and achieves results comparable to, though slightly worse than, MemoNet and EqMotion on the ETH-UCY dataset. On the SDD dataset, our method outperforms FlowChain while performing comparably to MemoNet. However, these existing methods are challenging to adapt to other tasks, such as trajectory imputation, because their designs are specifically tailored for prediction and are not well-suited for broader applications. Additionally, our focus is on modeling players in the sports domain, which differs from the pedestrian scenarios these methods typically address.

For the imputation and recovery tasks, we conducted experiments on the recently open-sourced time-series imputation dataset Traffic-Guangzhou (Chen et al., 2018), comparing our method with the latest imputation approaches, CSBI (Chen et al., 2023c) and BayOTIDE Fang et al. (2024). The results, presented in Table 12, show that our method outperforms CSBI in both RMSE and MAE and achieves results comparable to, though slightly worse than, BayOTIDE. One possible reason for this is that time-series datasets, unlike multi-agent datasets, lack the dense, structured interactions present among sports players. Additionally, these baselines are difficult to adapt to trajectory prediction tasks. Overall, these experiments further validate the generalizability and robustness of our proposed method.

**Additional Qualitative Examples** We present another visualization example from the Basketball-U dataset in Figure 5, to further demonstrate the effectiveness of our method. We can observe that the trajectories generated by our method are also closer to the ground truth and yield more accurate predictions for missing values. In addition, some baseline methods generate trajectories that are outside the basketball court boundaries, whereas our trajectories remain well-contained within the court.

Table 10: We compare our UniTraj with trajectory prediction methods on the ETH-UCY dataset. The best results are highlighted and the second best results are underlined.

| Method | $minADE_{20}/minFDE_{20} \downarrow$ | | | | | |
|---|---|---|---|---|---|---|
| | Eth | Hotel | Univ | Zara1 | Zara2 | Average |
| MemoNet (2022b) | **0.40/0.61** | **0.11/0.17** | 0.24/**0.43** | **0.18/0.32** | 0.14/0.24 | **0.21/0.35** |
| FlowChain (2023) | 0.55/0.99 | 0.20/0.35 | 0.29/0.54 | 0.22/0.40 | 0.20/0.34 | 0.29/0.52 |
| EqMotion (2023a) | **0.40/0.61** | 0.12/0.18 | **0.23/0.43** | **0.18/0.32** | **0.13/0.23** | **0.21/0.35** |
| **UniTraj (Ours)** | 0.43/0.62 | 0.13/0.19 | 0.25/**0.43** | 0.20/0.33 | 0.16/0.24 | 0.23/0.36 |

Table 11: We compare our UniTraj with trajectory prediction methods on the SDD dataset. The best results are highlighted and the second best results are underlined.

| Method | SDD |
|---|---|
| | $minADE_{20}/minFDE_{20} \downarrow$ |
| MemoNet (2022b) | **8.56/12.66** |
| FlowChain (2023) | 9.93/17.17 |
| **UniTraj (Ours)** | 8.68/12.78 |

Table 12: We compare our UniTraj with time-series imputation methods on the Traffic-Guangzhou dataset. The best results are highlighted and the second best results are underlined.

| Method | Traffic-GuangZhou | |
|---|---|---|
| | $RMSE \downarrow$ | $MAE \downarrow$ |
| CSBI | 4.790 | 3.182 |
| BayOTIDE | **3.820** | **2.687** |
| **UniTraj (Ours)** | 3.942 | 2.784 |

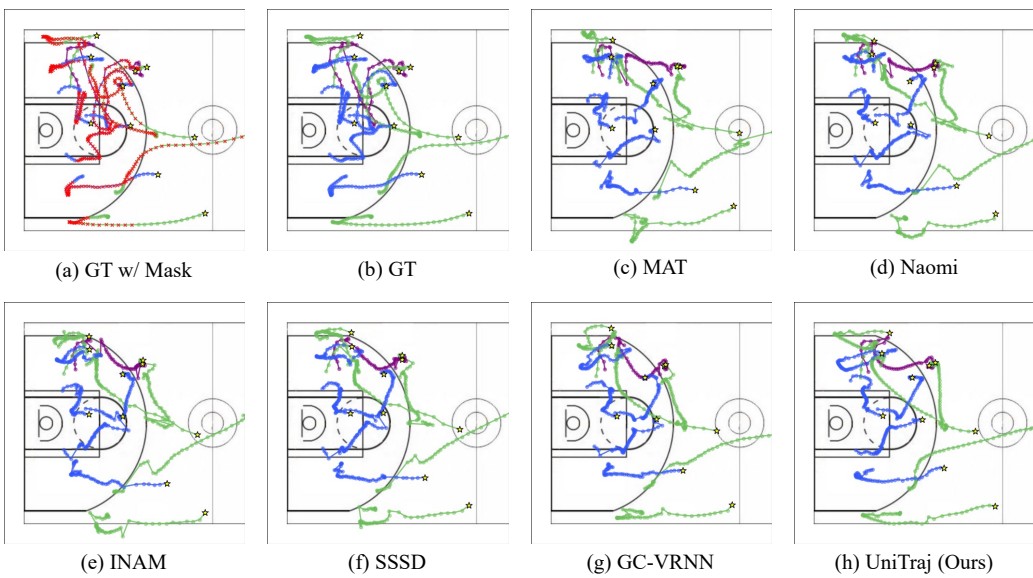

(a) GT w/ Mask    (b) GT    (c) MAT    (d) Naomi

(e) INAM    (f) SSSD    (g) GC-VRNN    (h) UniTraj (Ours)

Figure 5: Qualitative comparison between advanced baselines and our method. The ball's trajectory is shown in purple, offensive players are in green, and defensive players are in blue. Red "x" marks indicate masked locations and the starting points of the trajectories are highlighted with yellow stars.

