# OpenReview forum: "Sports-Traj: A Unified Trajectory Generation Model for Multi-Agent Movement in Sports"
_ICLR.cc/2025/Conference — ICLR 2025 Poster_

### Official Review · Reviewer_uin8 · 2024-10-24

**Soundness:** 2
**Presentation:** 3
**Contribution:** 3
**Rating:** 6
**Confidence:** 3

**Summary:**

The contributions of the paper are as follows:
(1) The authors unified trajectory prediction, imputation, and spatial-temporal recovery into a single framework.
(2) They introduced the UniTraj model, capable of processing arbitrary trajectories as masked inputs, making it adaptable to diverse and incomplete datasets, particularly useful in sports scenarios.
(3) The paper shows how the Ghost Spatial Masking (GSM) module and the Bidirectional Temporal Scaled (BTS) module help achieve state-of-the-art performance by improving spatial feature extraction and preserving temporal relationships in trajectory data.
(4) The authors curated and benchmarked three practical sports datasets, namely Basketball-U, Football-U, and Soccer-U, which will serve as valuable resources for other researchers working in the field of sports analytics.

**Strengths:**

The paper presents an original idea by unifying three tasks—trajectory prediction, imputation, and spatial-temporal recovery—into a single framework. The selection of network components, such as the Ghost Spatial Masking (GSM) module and the Bidirectional Temporal Scaled (BTS) module, is well justified, and their effectiveness is thoroughly evaluated through extensive experiments. The model's performance is further validated using three distinct sports datasets, demonstrating its robustness and applicability across different scenarios.

**Weaknesses:**

In real-world scenarios, trajectory masking isn't limited to binary values (0 or 1); detection errors often lead to incorrect trajectories. For instance, a referee might be mistakenly tracked as a player, or an incorrect ball-like object could be detected as the ball instead of the actual one. These errors highlight the need for models to handle not only missing but also erroneous data in trajectory prediction and recovery tasks.

**Questions:**

(1) At line 474, should it read "The 'w/o BTS' variant excludes the Bidirectional Temporal Scaled (BTS) module" instead of "w/o GSM"?

(2) At line 413, why is a total of K=20 trajectories generated? What is the variability among these 20 trajectories? I understand that generating multiple trajectories is necessary due to the model being generative, but I would like to know if selecting different values for K affects the experimental results (such as minADE). Additionally, were the baseline models evaluated under similar conditions, such as using K=20 trajectories?

(3) In Table 2, the minADE for UniTraj is reported as 94.59 pixels. What is the size of the soccer field in pixels? I want to understand how significant this error is in real-world measurements. Also, will you provide visual examples to demonstrate the quality of the generated trajectories, particularly for average cases?

**Details Of Ethics Concerns:**

No.

---

> ### Author Response · Authors · 2024-11-23
> **Response to Reviewer uin8 (Part 1/3)**
>
> **Dear Reviewer uin8**,
>
> ***We sincerely appreciate your recognition of our contributions and your constructive suggestions to improve our manuscript.***
> Below, we provide detailed responses to address your concerns.
>
> **[Weakness] Real-world Scenarios**
>
> **[A0]** That's a really insightful point, and we agree that addressing problems with erroneous data is a highly valuable direction, especially in the sports domain. While our current work focuses on a relatively simpler problem, we believe it still holds significant value for sports analysis.
> We have some initial thoughts on addressing the erroneous data scenario:
> 1. We could develop a probabilistic model to represent the erroneous data and incorporate approaches to quantify the uncertainty in the locations of the agents.
> 2. Beyond generating trajectories, we could also output the probabilistic distributions and uncertainty measures for these generated locations, which could provide deeper insights for sports analysis.
>
> We will include these points in our final version to better highlight the potential and importance of this problem.
>
> **[Q1] Typo**
>
> **[A1]** Thank you for your careful review. That's a typo, it should be "w/o BTS" instead. We have corrected it in the updated submission on Line 474, highlighted in blue. We will thoroughly review and polish the entire draft twice before the final version.

---

> ### Author Response · Authors · 2024-11-23
> **Response to Reviewer uin8 (Part 2/3)**
>
> **[Q2] Sampling K Trajectories**
>
> **[A2]** That’s a very insightful question. We follow pioneering trajectory prediction works[1][2] in setting K=20 for multiple trajectory generation to account for inherent multimodality, where multiple plausible paths can exist. Based on your suggestion, we evaluated our trained models with different values of K (K=10, K=20, K=30) and reported the mean and standard deviation (std) of all metrics across three datasets. **UniTraj*** indicates that the results are presented as mean±std.
>
> |Basketball   | |              |            |           |            |  |
> |------------|------------|:---------------------|:-------------------|:----------------|:-----------------------|:------------------------|
> |  **Method**  | **K**|**minADE$_{20}$**         | **OOB**          | **Step**       | **Path-L**    | **Path-D**     |
> | UniTraj |K=10 | 4.7667             | 6.11e-04          | 0.27           | 34.25                 | 241.06                 |
> | **UniTraj*** |K=10 | 4.7671±2.09e-05 | 6.10e-04±1.39e-06 | 0.27±8.61e-06 | 34.25±3.62e-04       | 240.95±0.22          |
> | UniTraj |K=20 | 4.7668             | 6.12e-04          | 0.27           | 34.25                 | 240.83                 |
> | **UniTraj***| K=20 | 4.7671±-1.74e-05 | 6.11e-04±2.17e-06 | 0.27±2.32e-06 | 34.25±3.50e-04       | 240.92±0.27          |
> | UniTraj |K=30 | 4.7666             | 6.08e-04          | 0.27           | 34.25                 | 240.84                 |
> | **UniTraj*** |K=30 | 4.7670±-1.56e-05 | 6.10e-04±1.84e-06 | 0.27±2.24e-06 | 34.25±3.65e-04       | 240.99±0.25          |
>
> |Football| |              |            |           |            |  |
> |------------|------------|:---------------------|:-------------------|:----------------|:-----------------------|:------------------------|
> |  **Method**  | **K**|**minADE$_{20}$**         | **OOB**          | **Step**       | **Path-L**    | **Path-D**     |
> | UniTraj       |K=10 | 3.5497              | 1.11e-04             | 0.23              | 19.27                  | 115.74                |
> | **UniTraj***      |K=10 | 3.5502 ± 5.84e-05   | 1.12e-04 ± 9.84e-07  | 0.23 ± 5.14e-05   | 19.27 ± 1.39e-03       | 116.39 ± 0.63         |
> | UniTraj       |K=20 | 3.5499              | 1.12e-04             | 0.23              | 19.26                  | 114.58                |
> | **UniTraj***      |K=20 | 3.5502 ± 5.90e-05   | 1.12e-04 ± 9.35e-07  | 0.23 ± 5.77e-05   | 19.27 ± 1.97e-03       | 115.88 ± 0.52         |
> | UniTraj       |K=30 | 3.5495              | 1.11e-04             | 0.23              | 19.27                  | 116.30                |
> | **UniTraj***      |K=30 | 3.5502 ± 6.99e-05   | 1.12e-04 ± 9.95e-07  | 0.23 ± 5.50e-05   | 19.27 ± 2.13e-03       | 115.79 ± 0.60         |
>
> |Soccer| |              |            |           |            |  |
> |------------|------------|:---------------------|:-------------------|:----------------|:-----------------------|:------------------------|
> |  **Method**  | **K**|**minADE$_{20}$**         | **OOB**          | **Step**       | **Path-L**    | **Path-D**     |
> | UniTraj      | K=10 |94.5840             | 3.65e-06             | 4.52              | 349.67                 | 2778.11                |
> | **UniTraj***     | K=10 |94.5994 ± 1.58e-03  | 3.48e-06 ± 1.66e-07  | 4.52 ± 1.20e-03   | 349.72 ± 0.08          | 2765.24 ± 24.59        |
> | UniTraj      | K=20 |94.5909             | 3.31e-06             | 4.52              | 349.73                 | 2805.79                |
> | **UniTraj***    | K=20 |94.5991 ± 1.53e-03  | 3.38e-06 ± 1.99e-07  | 4.52 ± 1.18e-03   | 349.68 ± 0.06          | 2753.72 ± 20.97        |
> | UniTraj      | K=30 |94.5778             | 3.65e-06             | 4.52              | 349.61                 | 2783.75                |
> | **UniTraj***   | K=30 |94.5994 ± 1.83e-03  | 3.41e-06 ± 2.12e-07  | 4.51 ± 1.32e-03   | 349.69 ± 0.07          | 2771.65 ± 18.72        |
>
> The results for all five metrics remain very similar across the three datasets with different values of K. In addition, both minADE$_{K}$
> and mean ADE are very close, with consistently low standard deviations.
>
> For the baselines, we implemented method MAT, Naomi, INAM, and SSSD for the deterministic generation, we implemented GC-VRNN with $K=20$ by sampling $Z$ from their prior distribution and reporting the minimum metrics.
>
> We will include all these results and further discuss this point in the Experiment Section. Additionally, we will provide supplementary implementation details for the baselines in the final version

---

> ### Author Response · Authors · 2024-11-23
> **Response to Reviewer uin8 (Part 3/3)**
>
> **[Q3] Soccer Metrics and Visualizations**
>
> **[A3]** According to the dataset[3], a soccer field measures 105 meters in length and 68 meters in width, mapped to a pixel resolution of 3840 x 2160. However, the image does not perfectly align with the soccer field dimensions, as noted in the dataset, and only a rough calculation can be made with some margin of error.
>
> · 94.59 pixels converted to the horizontal direction (length): approximately 2.59 meters
>
> · 94.59 pixels converted to the vertical direction (width): approximately 2.98 meters
>
> We have included two visualization examples from the Basketball dataset, along with analysis, in the **Appendix** of our **updated submission**. The results show that the trajectories generated by our method are more accurate and smoother compared to the baselines, further validating the effectiveness of our proposed approach. We will include more qualitative results in the final version.
>
> [1] Social-Gan: Socially Acceptable Trajectories with Generative Adversarial Networks. CVPR 2018.
> [2] Trajectron++: Dynamically-Feasible Trajectory Forecasting with Heterogeneous Data. ECCV 2020.
> [3] SoccerTrack: A Dataset and Tracking Algorithm for Soccer with Fish-eye and Drone Videos. CVPR 2022.
>
> ***We sincerely appreciate your valuable suggestions and insightful comments. We hope our response effectively addresses your concerns.***

---

> ### Comment · Reviewer_uin8 · 2024-11-26
> **Please provide more concrete details on addressing the issue of erroneous data in real-world scenarios.**
>
> Thank you for sharing your initial thoughts on addressing the issue of erroneous data in real-world scenarios. The proposed directions, such as developing a probabilistic model for representing errors and incorporating uncertainty measures into trajectory generation, are intriguing and could significantly enhance the applicability of your work. Please provide more concrete details on these ideas, as they would strengthen the paper and demonstrate the potential of your approach to handle real-world challenges effectively.

---

> > ### Author Response · Authors · 2024-11-26
> > **Details and Insights of Addressing the Issue of Erroneous Data**
> >
> > Thank you for your encouraging feedback. We greatly appreciate your interest in our proposed directions and agree that providing more concrete details will strengthen the paper’s contribution. Below, we elaborate on our initial ideas:
> >
> > * **Data Representation and Simulation**
> >
> > In our current work, we use a binary mask (0 or 1), which does not fully capture the complexity of erroneous data. To better mimic real-world scenarios, we propose incorporating pre-trained detection models with images/videos as input to generate detection results for the ball and players. These detection results, while realistic, may inherently introduce errors such as misidentifications (e.g., detecting referees as players or false ball detections). This approach would enable us to build a dataset that better reflects real-world conditions and serves as a foundation for developing robust trajectory generation models.
> >
> > * **Probabilistic Modeling and Uncertainty Quantification**
> >
> > Building on the enriched dataset, we propose extending our current framework by using a Gaussian Mixture Model (GMM) instead of a single Gaussian distribution to represent the trajectory distribution (as in Equation 9 of our work).
> >
> > 1. GMM Parameters: The model would output both the parameters of each Gaussian component and the corresponding weights $p_k$​, representing the probability of each component. For example, we can set K=20 to align with the complexity of multi-modal trajectories.
> >
> > 2. Training: We can retain the Winner-Take-All loss to minimize the distance between the best Gaussian component and the ground truth while adding a cross-entropy loss to maximize the probability of the selected component.
> >
> > 3. Inference: During inference, this setup provides not only generated trajectories but also their associated confidence levels (uncertainties), offering deeper insights into trajectory reliability.
> >
> > * **Pre-Filtering Module**
> > Another potential extension involves integrating a pre-filtering module, such as an anomaly detection approach. This module could classify detected agents and objects into normal and abnormal categories, thereby mitigating error propagation into the trajectory generation process.
> >
> > While these ideas extend beyond the scope of our current work, they represent promising pathways to address real-world challenges effectively. We will add a dedicated discussion section to outline these future directions, emphasizing their importance and potential impact on the domain in our final version.
> >
> > We hope this additional detail clarifies our vision and demonstrates our commitment to advancing the applicability of our approach. Thank you again for your insightful comments.

---

> > ### Author Response · Authors · 2024-12-03
> >
> > Dear Reviewer uin8,
> >
> > Thank you once again for your valuable feedback. As the discussion deadline is approaching, if you have any further questions or concerns, we would be more than happy to address them.
> >
> > Best regards,
> >
> > Authors of Submission 4628

---

### Official Review · Reviewer_ZPa2 · 2024-10-25

**Soundness:** 2
**Presentation:** 4
**Contribution:** 3
**Rating:** 6
**Confidence:** 3

**Summary:**

They focus on the domain of sports and address the problem of modeling multi-agent trajectories by considering various situations in real practice, emphasizing the need for a general approach. To accommodate diverse real-world scenarios, they introduce a unified trajectory generation task that simultaneously handles multiple input situations.

**Strengths:**

This paper introduces a new problem (a unified trajectory generation task) along with several new datasets, with relatively comprehensive experiments and a substantial amount of work. They propose a Unifed lrajectoryenyironments.Generation model, UniTraj, that processes arbitrary trajectories as masked inputs.adaptable to diverse scenarios in the domain of sports games. They further extend recent State Space models (SSMs), known as the Mamba model, into a Bidirectional Temporal Mamba (BTM) to better capture temporal dependencies.

**Weaknesses:**

1.Could you please try LSTM/CNN or any other backbones instead of MLP to achieve better performance?

2.Some equations are very similar, so you can combine them together (such as Eq.7). It's unnecessary to write them again.

3.You can also compare the FLOPs and total parameters with state-of-the-art methods to demonstrate your models’ efficiency.

**Questions:**

See weakness above.

**Details Of Ethics Concerns:**

No.

---

> ### Author Response · Authors · 2024-11-20
> **Response to Reviewer ZPa2**
>
> **Dear Reviewer ZPa2**,
>
> ***We sincerely appreciate your recognition of our contributions and your constructive suggestions to improve our manuscript.***
> Below, we provide detailed responses to address your concerns.
>
> **[Q1] LSTM/CNN replace MLP**
>
> **[A1]** If we understand correctly, your suggestion is to replace the MLP decoder with LSTM or CNN, as mentioned in the limitation section. Following your suggestion, we implemented more powerful networks, including LSTM and CNN decoders, to replace the MLP decoder. The results are as follows:
> |Basketball   | |              |            |           |            |
> |:----------|:-------------------:|:------------:|:----------:|:---------:|:----------:|
> |  **Method**  | **minADE$_{20}$**         | **OOB**          | **Step**       | **Path-L**    | **Path-D**     |
> | UniTraj    | 4.77                | 6.12e-04     | 0.27       | 34.25     | 240.83     |
> | w/ LSTM    | 4.54                | 0            | 0.18       | 32.03     | 135.40     |
> | w/ CNN     | 4.62                | 5.57e-05     | 0.17       | 30.86     | 126.54     |
>
> |Football       | |              |            |           |            |
> |:----------|:-------------------:|:------------:|:----------:|:---------:|:----------:|
> |  **Method**  | **minADE$_{20}$**         | **OOB**          | **Step**       | **Path-L**    | **Path-D**     |
> | UniTraj    | 3.55                | 1.12e-04     | 0.23       | 19.26     | 114.58     |
> | w/ LSTM    | 2.93                | 0            | 0.15       | 15.03     | 81.97      |
> | w/ CNN     | 3.32                | 8.87e-05     | 0.14       | 15.57     | 77.88      |
>
> |Soccer| |              |            |           |            |
> |:----------|:-------------------:|:------------:|:----------:|:---------:|:----------:|
> |  **Method**  | **minADE$_{20}$**         | **OOB**          | **Step**       | **Path-L**    | **Path-D**     |
> | UniTraj    | 94.59               | 3.31e-06     | 4.52       | 349.73    | 2805.79    |
> | w/ LSTM    | 77.90               | 0            | 2.77       | 223.54    | 1195.83    |
> | w/ CNN     | 92.13               | 3.91e-05     | 2.79       | 244.08    | 1065.09    |
>
> For the w/ LSTM variant, we replace the MLP decoder with a single-layer LSTM with a hidden dimension of 128. For the w/ CNN variant, we replace the MLP decoder with a three-layer Conv2D network, where the first two layers use 64 filters with a kernel size of (3, 1), and the last layer uses 64 filters with a kernel size of (1, 1).
>
> Our results show that using more powerful decoders improves performance across all three datasets. Among the tested decoders, LSTM outperforms CNN, as it better captures temporal dependencies between different time steps. However, since the primary contribution of our work is introducing the new trajectory generation setting, we did not devote too much to optimizing the module designs. We hope that our work inspires the community to develop more advanced network architectures for this task. We'll add those discussions in the final version.
>
> **[Q2] Equations**
>
> **[A2]** Thank you for your careful review. We revise Eq. 7 as follows:
> $$\overset{\leftrightarrow}{F}_{bts}= 1/\exp(\varphi_s(\overset{\leftrightarrow}{S};\mathbf{W}_s))$$
> We will double-check and further polish the draft before the final version.
>
> **[Q3] Params and FLOPs**
>
> **[A3]** Thank you for your insightful comments. Below are the results of the number of parameters and GFLOPs on the Basketball dataset, compared with advanced baselines.
>
> | Method   | #Params   | GFLOPs |
> |:----------|:-----------:|:--------:|
> | MAT      | 9.21M     | 0.39   |
> | Naomi    | $\underline{2.21}$M    | **0.21**   |
> | INAM     | 2.40M     | 0.52   |
> | SSSD     | 48.37M    | 0.99   |
> | GC-VRNN  | 2.76M     | 0.48   |
> | UniTraj  | **1.77M**     | $\underline{0.33}$   |
>
> Our method has a total of 1.77M model parameters and 0.33 GFLOPs. Among all the advanced baselines, our model has the smallest number of parameters and the second lowest FLOPs, slightly higher than the baseline Naomi. This is because Naomi's backbone is an RNN, which requires fewer computations compared to our Transformer and Mamba structure. These results validate the efficiency of our method. We'll add those results in the final version.
>
>
> ***We sincerely appreciate your valuable suggestions and insightful comments. We hope our response effectively addresses your concerns.***

---

> > ### Comment · Reviewer_ZPa2 · 2024-11-23
> > **Response**
> >
> > Thank you so much. Your response has addressed my concerns.

---

> ### Author Response · Authors · 2024-11-23
> **Further Response to Reviewer ZPa2**
>
> Dear Reviewer ZPa2,
>
> Thank you for helping us improve our paper so far. We are very glad that we've addressed your concern. May we know if you have further questions? If not, would you consider increasing the score as a recognition of our rebuttal effort so far? Thank you so much!
>
> Best regards,
>
> Authors of Submission 4628

---

### Official Review · Reviewer_HqF9 · 2024-10-26

**Soundness:** 3
**Presentation:** 3
**Contribution:** 3
**Rating:** 6
**Confidence:** 4

**Summary:**

This paper proposed a Unified Trajectory Generation model, UniTraj, that processes arbitrary trajectories as masked inputs, adaptable to diverse scenarios in the domain of sports games, integrating trajectory prediction, imputation, and spatial-temporal recovery into a single framework. Key contributions include the development of a Ghost Spatial Masking module for spatial feature extraction, a Bidirectional Temporal Mamba encoder for enhanced temporal modeling, and the curation of three new sports datasets for robust evaluation.

**Strengths:**

originality: This paper proposes a novel approach for handling multiple tasks, including trajectory prediction, imputation, and spatial-temporal recovery for multi-agent movement analysis. It introduces the innovative Ghost Spatial Masking module and extends the Mamba model with a new Bidirectional Temporal Scaled module to enhance the extraction of comprehensive spatial-temporal features from various incomplete trajectory inputs.

quality: The quality of this is generelly good. The method is well explained.

clarity: The paper is well-written and organized logically. The use of figures, especially the architectural diagrams and flowcharts, effectively aids in understanding the model's components and their interactions. Each section of the paper builds upon the previous one, leading to a cohesive narrative from problem formulation to experimental validation.

significance: The significance of this research lies in its potential impact on sports analytics and related fields requiring accurate multi-agent trajectory analysis.

**Weaknesses:**

The research problem is not well explained. The paper does not explain why all-in-one methods for the three tasks, trajectory prediction, imputation, and spatial-temporal recovery, have not been proposed. What are the challenges? The proposed methods should be compared separately with relevant methods for each task. Although the paper tests the model on three sports datasets, sports differ significantly in their dynamics and player interactions. More varied tests are needed.

**Questions:**

1.  Why have all-in-one methods for the three tasks, trajectory prediction, imputation, and spatial-temporal recovery, not been proposed.?
2. What are the challenges?
3. Comparisons with other methods should be given.

---

> ### Author Response · Authors · 2024-11-23
> **Response to Reviewer HqF9  (Part 1/2)**
>
> Dear Reviewer HqF9,
>
> ***We sincerely appreciate your recognition of our contributions and your constructive suggestions to improve our manuscript.***
> We provide the following detailed responses to address your concerns.
>
> **[Q1] Motivation**
>
> **[A1]** That’s a good question. The three tasks, trajectory prediction, imputation, and spatial-temporal recovery, are related but different, with differences in input formats, objectives, and methodologies. For example, trajectory prediction focuses on forecasting future values, imputation deals with filling in missing data, and spatial-temporal recovery simultaneously reconstructs both spatial and temporal patterns. In real-world applications such as sports analytics, these scenarios often occur together, creating a strong motivation for an all-in-one approach. A unified framework would seamlessly address these overlapping demands, reducing the need for separate pipelines and improving overall efficiency. While some related methods [1][2] attempt to tackle prediction and imputation simultaneously, their masking strategies fall short in handling diverse missing patterns. Empirical results further demonstrate the superiority of our proposed method.
>
> We will enhance the introduction section in our final version to further strengthen and clarify the motivation.
>
>
>
>
>
>
> **[Q2] Challenges**
>
> **[A2]** The challenges of developing an all-in-one method include the following:
>
> 1. Different missing data patterns: Trajectory prediction requires extrapolation of future values, imputation often deals with random missing data, and spatial-temporal recovery involves complex dependencies in both space and time.
>
> 2. Balancing context and forecasting: A unified model must balance local context modeling for imputation and spatial-temporal recovery with long-term forecasting for trajectory prediction, which is inherently challenging.
>
> 3. Dynamic interaction modeling: Player interactions in sports vary significantly across datasets, requiring the model to generalize across diverse scenarios.
>
> These challenges highlight why such an all-in-one approach has not been extensively explored and underscore the contributions of our work in addressing these issues. We will incorporate these discussions into the final version to emphasize the challenges.

---

> ### Author Response · Authors · 2024-11-23
> **Response to Reviewer HqF9 (Part 2/2)**
>
> **[Q3] Comparison**
>
> **[A3]** Thank you for your valuable suggestions. We conducted experiments to separately compare our method with relevant approaches for each task.
>
> For the prediction task, we evaluated our method on the pedestrian datasets ETH-UCY and SDD. The results are shown as follows:
>
> ETH-UCY|ADE/FDE (K=20)|      |        |      |      |      |
> |---------------|:-----------:|:-----------:|:-----------:|:-----------:|:-----------:|:-----------:|
> | **Method**        | **ETH**       | **Hotel**     | **Univ**      | **Zara1**     | **Zara2**     | **Average**   |
> | FlowChain[5] | 0.55/0.99 | 0.20/0.35 | 0.29/0.54 | 0.22/0.40 | 0.20/0.34 | 0.29/0.52 |
> | MemoNet[4]   | 0.40/0.61 | 0.11/0.17 | 0.24/0.43 | 0.18/0.32 | 0.14/0.24 | 0.21/0.35 |
> | EqMotion[3]  | 0.40/0.61 | 0.12/0.18 | 0.23/0.43 | 0.18/0.32 | 0.13/0.23 | 0.21/0.35 |
> | UniTraj       | 0.43/0.62 | 0.13/0.19 | 0.25/0.43 | 0.20/0.33 | 0.16/0.24 | 0.23/0.36 |
>
> SDD ||
> |---------------|:-----------:|
> | **Method**        | **ADE/FDE (K=20)**|
> | FlowChain[5] | 9.93/17.17 |
> | MemoNet[4]   | 8.56/12.66 |
> | UniTraj       | 8.68/12.78 |
>
> Our method outperforms FlowChain [5] and achieves results comparable to, though slightly worse than, the state-of-the-art baselines MemoNet[3] and EqMotion[4]. However, these methods are challenging to adapt to other tasks, such as trajectory imputation, because their designs are specifically tailored for prediction tasks and are not well-suited for broader applications. Additionally, our focus is on modeling players within the sports domain, which differs from the pedestrian scenarios addressed by these methods.
>
> For the imputation and recovery tasks, we conducted experiments on the recently open-sourced time-series imputation dataset Traffic-Guangzhou, comparing our method with the latest imputation approaches, CSBI[6] and BayOTIDE[7]. The results are shown as follows:
>
> Traffic-GuangZhou|||
> |---------------|:-----------:|:-----------:|
> | **Method**        | **RMSE**| **MAE**|
> |CSBI[6] | 4.790 | 3.182|
> | BayOTIDE[7] | 3.820 | 2.687|
> | UniTraj       | 3.942 | 2.784|
>
> Our method outperforms CSBI[6] in both RMSE and MAE and achieves results comparable to, though slightly worse than, BayOTIDE[7]. One reason for this is that time-series datasets, unlike multi-agent datasets, lack the dense and structured interactions present among sports players. Additionally, these baselines are challenging to adapt to the trajectory prediction task.
>
> In our submission, the baselines we compared, such as INAM[8] and GC-VRNN[1], were originally proposed for joint imputation and prediction tasks, and the empirical results demonstrate the superiority of our method. We will include these experiments and provide additional discussions in our final version to further showcase the effectiveness and generalizability of our proposed approach.
>
> [1] Uncovering the Missing Pattern: Unified Framework Towards Trajectory Imputation and Prediction. CVPR 2023.
> [2] Multiple-Level Point Embedding for Solving Human Trajectory Imputation with Prediction. TSAS 2023.
> [3] EqMotion: Equivariant Multi-Agent Motion Prediction with Invariant Interaction Reasoning. CVPR 2023.
> [4] Remember Intentions: Retrospective-Memory-based Trajectory Prediction. CVPR 2022.
> [5] Fast Inference and Update of Probabilistic Density Estimation on Trajectory Prediction. ICCV 2023.
> [6] Provably Convergent Schrödinger Bridge with Applications to Probabilistic Time Series Imputation. ICML 2023.
> [7] BayOTIDE: Bayesian Online Multivariate Time series Imputation with functional decomposition.ICML 2024.
> [8] Imitative Non-Autoregressive Modeling for Trajectory Forecasting and Imputation. CVPR 2020.
>
>
> ***We sincerely appreciate your valuable suggestions and insightful comments.*** We hope our response effectively addresses your concerns.

---

> > ### Comment · Reviewer_HqF9 · 2024-11-24
> >
> > Thanks for your efforts to address my concerns.

---

> > > ### Author Response · Authors · 2024-11-24
> > > **Further Response to Reviewer HqF9**
> > >
> > > Dear Reviewer HqF9,
> > >
> > > Thank you for your valuable comments, which have greatly helped us improve our paper. We are very glad to address your concerns and would be happy to answer any further questions you might have. If there are no additional questions, we kindly ask if you might consider increasing the score as recognition of our rebuttal efforts. Thank you so much!
> > >
> > > Best regards,
> > >
> > > Authors of Submission 4628

---

### Official Review · Reviewer_YWxk · 2024-11-03

**Soundness:** 2
**Presentation:** 3
**Contribution:** 2
**Rating:** 6
**Confidence:** 4

**Summary:**

The authors focus on the trajectory prediction in sport scenes. They propose a trajectory generation model. They extend the Mamba model into a bidirectional temporal Mamba for the purpose to enhance the temporal dependencies with a transformer encoder for feature extractor. The proposed method aims to. solve the trajectory prediction, imputation and spatial-temporal recovery tasks in a unified paradigm. The applied methods are designed specified to the sports scenarios but some proposed modules can be appliable to more general tasks. The authors construct several sport scenario focused datasets with existing datasets. On the proposed benchmarking platform, the proposed method achieves good experimental results.

**Strengths:**

- the proposed method is well designed for the sport scenes, such as soccer or basketball. With also tracking and forecasting the ball motion, the proposed method can be useful for the related sport activities..
- The proposed ghost masking embedding used to replace the usual head token can provide the order-invariant properties to the agent features, which can be extended to many related applications without losing generalization.
- Built upon the usual Mamba blocks, the proposed Bidirectional Temporal Mamba provides an improved fashion of processing spatial-temporal features with good adaptation to handle features on missing time steps.
- On the sports datasets, Basketball, Soccer and Football videos, the proposed method shows good quantitative results when compared to other related works.
- Overall, the paper is well written and the technical section can be easily followed.

**Weaknesses:**

1. My main concern about the proposed method is its generalizability. To be precise, the proposed method is evaluated on the three datasets Basketball-U, Soccer-U and Football-U, which are built by the authors themselves thus lacking a well established benchmarking. Therefore, it is hard to estimate the significance of the experimental performance by the provided benchmarking results. I would suggest the authors to add the experiments on the existing benchmarks, such as SDD, ETH/UCY or HM3.6M and include the recently published trajectory prediction methods into the comparison for a more convincing and well-established benchmarking.
2. The proposed methods model the ball trajectory and offensive/defensive player position explicitly. It is not clear whether the other baselines also follow this convention. For many previous works on NBA benchmark, based on which Basketball-U dataset is constructed, only the players position is considered. Such details of implementation alignment between the proposed method and baseline methods included in the benchmark results is critical to conduct a fair experiment and provide reliable experimental evidence.
3. There are many more recent related works, though they are mostly published on the benchmarking of ETH/UCY and SDD, should have been included in the experimental comparison, to provide a up-to-date evaluation of the quantitative significance of the proposed method. To name some: Eqmotion[1], MemoiNet[2], Flowchain[3].

Reference:

[1] “**EqMotion: Equivariant Multi-Agent Motion Prediction with Invariant Interaction Reasoning”, CVPR 2023**

[2] “**Remember Intentions: Retrospective-Memory-based Trajectory Prediction”, CVPR 2022**

[3 “**Fast Inference and Update of Probabilistic Density Estimation on Trajectory Prediction”, ICCV 2023**

**Questions:**

My questions to be answered and my concerns to be addressed have been discussed in the `weakness` section.

---

> ### Author Response · Authors · 2024-11-23
> **Response to Reviewer YWxk (Part 1/3)**
>
> Dear Reviewer YWxk,
>
> ***Thank you so much for your valuable suggestions and detailed comments.***
> We provide the following detailed responses to address your concerns.
>
> **[Q1] Generalizability**
>
> **[A1]** Thank you for your constructive comment. To assess the generalizability of our method, we followed your suggestions and conducted experiments on the ETH/UCY and SDD datasets for the trajectory prediction task. The results of stochastic predictions with K=20 are shown below:
>
> ETH-UCY|ADE/FDE (K=20)|      |        |      |      |      |
> |---------------|:-----------:|:-----------:|:-----------:|:-----------:|:-----------:|:-----------:|
> | **Method**        | **ETH**       | **Hotel**     | **Univ**      | **Zara1**     | **Zara2**     | **Average**   |
> | FlowChain[3] | 0.55/0.99 | 0.20/0.35 | 0.29/0.54 | 0.22/0.40 | 0.20/0.34 | 0.29/0.52 |
> | MemoNet[2]   | 0.40/0.61 | 0.11/0.17 | 0.24/0.43 | 0.18/0.32 | 0.14/0.24 | 0.21/0.35 |
> | EqMotion[1]  | 0.40/0.61 | 0.12/0.18 | 0.23/0.43 | 0.18/0.32 | 0.13/0.23 | 0.21/0.35 |
> | UniTraj       | 0.43/0.62 | 0.13/0.19 | 0.25/0.43 | 0.20/0.33 | 0.16/0.24 | 0.23/0.36 |
>
> SDD ||
> |---------------|:-----------:|
> | **Method**        | **ADE/FDE (K=20)**|
> | FlowChain[3] | 9.93/17.17 |
> | MemoNet[2]   | 8.56/12.66 |
> | UniTraj       | 8.68/12.78 |
>
> Our method demonstrates better performance than FlowChain[3] and achieves results comparable to, though slightly worse than, the state-of-the-art baselines MemoNet[2] and EqMotion[3].
>
> One reason is that our modules are specifically designed for sports datasets, which feature more structured interactions among players, whereas pedestrian movement patterns tend to be more casual and random. Despite this difference, our results show that the proposed modules effectively capture spatial-temporal features from observed trajectories, further validating the generalizability of our approach.
>
> Additionally, our method is applicable to other trajectory-relevant tasks. In the final version, we will include more baseline comparisons for trajectory prediction to further emphasize the generalizability of our method.

---

> > ### Comment · Reviewer_YWxk · 2024-11-24
> >
> > The new experiments on ETH/UCY and SDD are helpful to understand the performance of the proposed method on usual trajectory prediction benchmarks. I will take the new experiments into consideration when rethinking the review rating.

---

> > > ### Author Response · Authors · 2024-11-24
> > > **Further Response to Reviewer YWxk**
> > >
> > > Dear Reviewer YWxk,
> > >
> > > Thank you for your valuable comments and recognition of our trajectory prediction experiments. Your feedback has greatly helped us improve our paper, and we are glad to address your concerns. We would be happy to answer any further questions you might have.
> > >
> > > Thank you so much for your thoughtful review!
> > >
> > >
> > >
> > > Best regards,
> > >
> > > Authors of Submission 4628

---

> > > > ### Comment · Reviewer_YWxk · 2024-11-29
> > > >
> > > > I appreciate the feedback extra experiments and discussion provided by the authors.
> > > >
> > > > In the original review, my main concern is about generalizability as the paper originally only applies experiments on the newly built datasets. However the authors provided extra experiments on the more canonical and standard benchmarks during the rebuttal. Though the performance is not very significant anymore, the extra experiments do relieve my concern about the generalizability.
> > > >
> > > > Also, the authors promised to add a discussion about recent related works, the lack of which made another part of my concern originally.
> > > >
> > > > Given the improvement from the authors above, I have adjusted my rating score to the paper.

---

> > > > > ### Author Response · Authors · 2024-11-30
> > > > >
> > > > > Thank you for your feedback and recognizing our work with an improved score!
> > > > >
> > > > > Your review has been incredibly helpful in improving the quality of our paper. We will definitely include the additional experiments and discussion of related works in our final version.
> > > > >
> > > > > If you have any further questions, we are more than happy to answer them.

---

> ### Author Response · Authors · 2024-11-23
> **Response to Reviewer YWxk (Part 2/3)**
>
> **[Q2] Ball and Category**
>
> **[A2]** That’s an insightful point. Following your suggestion, we conducted three experiments:
> (1) keeping the ball while removing the offensive/defensive player categories,
> (2) removing the ball while keeping and concatenating the offensive/defensive player categories,
> (3) removing both the ball and the offensive/defensive player categories.
> The results are shown in the following tables.
>
> | Basketball | Variant |        |         |         |         |         |         |         |
> |--------------------|:-----------:|:--------------:|:--------------:|:-------------:|:-----------:|:------:|:--------:|:---------:|
> Method    | Ball         | Category     | minADE$_{20}$ | OOB       | Step | Path-L | Path-D  |
> | UniTraj   | $\checkmark$  | $\times$   | 4.80        | 6.53e-04  | 0.29 | 35.23  | 143.05  |
> | UniTraj   | $\times$ | $\checkmark$    | 4.67        | 4.51e-04  | 0.27 | 31.95  | 113.17  |
> | UniTraj   | $\times$ | $\times$  | 4.65        | 3.59e-04  | 0.26 | 32.56  | 131.67  |
> | UniTraj  (Ours) |  $\checkmark$ |  $\checkmark$ | 4.77        | 6.12e-04  | 0.27 | 34.25  | 240.83  |
>
>
> | Football| Variant |        |         |         |         |         |         |         |
> |--------------------|:-----------:|:--------------:|:--------------:|:-------------:|:-----------:|:------:|:--------:|:---------:|
> Method    | Ball         | Category     | minADE$_{20}$ | OOB       | Step | Path-L | Path-D  |
> | UniTraj   | $\checkmark$  | $\times$   | 3.60         | 1.56e-04  | 0.24 | 19.34  | 138.99  |
> | UniTraj   | $\times$ | $\checkmark$    | 3.38         | 1.30e-04  | 0.23 | 18.85  | 111.43  |
> | UniTraj   | $\times$ | $\times$  |3.41         | 8.74e-05  | 0.24 | 19.13  | 122.83  |
> | UniTraj  (Ours) |  $\checkmark$ |  $\checkmark$ | 3.55         | 1.12e-04  | 0.23 | 19.26  | 114.58  |
>
>
> | Soccer| Variant |        |         |         |         |         |         |         |
> |--------------------|:-----------:|:--------------:|:--------------:|:-------------:|:-----------:|:------:|:--------:|:---------:|
> Method    | Ball         | Category     | minADE$_{20}$ | OOB       | Step | Path-L | Path-D  |
> | UniTraj   | $\checkmark$  | $\times$   |  96.09        | 1.66e-06  | 4.44 | 336.14 | 9146.07  |
> | UniTraj   | $\times$ | $\checkmark$    | 92.69        | 3.43e-07  | 3.70 | 289.46 | 1343.07  |
> | UniTraj   | $\times$ | $\times$  | 89.37        | 6.85e-07  | 4.03 | 304.41 | 2189.38  |
> | UniTraj  (Ours) |  $\checkmark$ |  $\checkmark$ | 94.59        | 3.31e-06  | 4.52 | 349.73 | 2805.79  |
>
> We observe that removing the category information leads to a performance drop, as this information plays an important role in sports. Interestingly, better performance is achieved when removing the ball. A potential reason is that the ball's trajectory is often unstable and influenced by external forces, introducing randomness and outliers that may disrupt the model's ability to learn overall movement patterns. The gap in dynamic characteristics between the ball and player trajectories impacts learning.
>
> Overall, our proposed method achieves the best performance when both the ball and category information are removed. We will include these results for a fair comparison and provide additional analysis in the final version.

---

> > ### Comment · Reviewer_YWxk · 2024-11-24
> >
> > I appreciate the experiments with the ablation of the ball and player category.
> >
> > Yes, this is truly interesting that removing ball trajectory from the method makes a better performance. The new experiments provide a more clear and stronger evidence about the experimental advantage of the proposed method on the sports datasets.

---

> > > ### Author Response · Authors · 2024-11-24
> > > **Further Evaluations on Removing Ball**
> > >
> > > Dear Reviewer YWxk,
> > >
> > > Thank you for your recognition of our efforts in conducting ablation experiments. To further explore and provide deeper insights into this interesting finding, we conducted additional evaluations by removing the ball.
> > >
> > > Specifically, we evaluated our trained models by separately assessing the metrics for “only players” and “only ball” across three datasets. The corresponding ground truth (GT) values are also in the following tables for reference.
> > >
> > > | Basketball Evaluation|        |              |         |         |         |         |         |
> > > |--------------------|-----------|:--------------:|:-------------:|:-----------:|:------:|:--------:|:---------:|
> > > |**Method**    | **Variant** | **minADE$_{20}$** | **OOB**       | **Step** | **Path-L** | **Path-D**  |
> > > | GT       | complete      | 0           | 0         | 0.17 | 37.61  | 269.49   |
> > > | UniTraj  | complete      | 4.77        | 6.12e-04  | 0.27 | 34.25  | 240.83   |
> > > | GT       | only players  | 0           | 0         | 0.12 | 34.13  | 261.86   |
> > > | UniTraj  | only players  | 4.57        | 6.53e-04  | 0.25 | 31.99  | 117.83   |
> > > |  GT       | only ball     | 0           | 0         | 0.68 | 72.36  | 261.86   |
> > > |  UniTraj  | only ball| 6.48        | 1.61e-04  | 0.52 | 56.82  | 233.53   |
> > >
> > > | Football Evaluation|        |              |         |         |         |         |         |
> > > |--------------------|-----------|:--------------:|:-------------:|:-----------:|:------:|:--------:|:---------:|
> > > |**Method**    | **Variant** | **minADE$_{20}$** | **OOB**       | **Step** | **Path-L** | **Path-D**  |
> > > | GT       | complete      | 0           | 0         | 0.03 | 12.56  | 76.68    |
> > > | UniTraj  | complete      | 3.55        | 1.12e-04  | 0.23 | 19.26  | 114.58   |
> > > | GT       | only players  | 0           | 0         | 0.02 | 11.95  | 49.73    |
> > > | UniTraj  | only players  | 3.47        | 5.94e-04  | 0.23 | 19.07  | 114.17   |
> > > | GT       | only ball     | 0           | 0         | 0.14 | 26.03  | 76.68    |
> > > | UniTraj  | only ball| 4.93        | 5.94e-04  | 0.27 | 23.69  | 115.11   |
> > >
> > > | Soccer Evaluation|        |              |         |         |         |         |         |
> > > |--------------------|:-----------|:--------------:|:-------------:|:-----------:|:------:|:--------:|:---------:|
> > > |**Method**    | **Variant** | **minADE$_{20}$** | **OOB**       | **Step** | **Path-L** | **Path-D**  |
> > > | GT       | complete      | 0           | 0         | 0.52 | 112.92  | 951.00   |
> > > | UniTraj  | complete      | 94.59       | 3.31e-06  | 4.52 | 349.73  | 2805.79  |
> > > | GT       | only players  | 0           | 0         | 0.52 | 105.82  | 951.00   |
> > > | UniTraj  | only players  | 87.25       | 3.43e-06  | 4.40 | 339.99  | 2724.56  |
> > > | GT       | only ball     | 0           | 0         | 0.40 | 269.00  | 922.33   |
> > > | UniTraj  | only ball| 218.71      | 0         | 7.17 | 557.76  | 2058.87  |
> > >
> > > We can observe that the evaluation results for minADE of "only ball" are much worse than those of "complete" and "only players".
> > >
> > > Looking into the ground truth (GT) of these variants, we find that in the "only ball" variant, the metrics Step, Path-L, and Path-D differ significantly from those of the "only players" variant in all three datasets. Specifically, Step measures the average change in step size, Path-L measures the average trajectory length, and Path-D measures the maximum difference in trajectory lengths. These differences indicate that the ball's movement is more dynamic and unstable, with a huge gap compared to players' movements. The ball's motion is often influenced by external forces, making it more challenging to predict than the players' movements.
> > >
> > > Another important aspect is the dataset itself: the number of ball trajectories is relatively smaller than that of player trajectories, adding an additional layer of difficulty. Therefore, developing a method that balances both the ball and players would be an interesting direction to explore. The ball’s movement, to some extent, reflects the players' actions or intentions during an offensive sequence, which is critical for real-world sports analysis and should not be overlooked.
> > >
> > > We will include these experiments and discussions in our final version to provide deeper insights into the sports analysis domain. Thank you once again for your valuable comments, which have inspired us to delve deeper into this topic and uncover additional findings.

---

> ### Author Response · Authors · 2024-11-23
> **Response to Reviewer YWxk (Part 3/3)**
>
> **[Q3] Prediction References**
>
> **[A3]** Thank you for providing those excellent papers relevant to our work. We have compared our results with them in our response **[A1]** and will also cite and discuss them in the related work section.
>
> Specifically, in[1], a prediction model named EqMotion is proposed, which integrates equivariant geometric and invariant pattern feature learning with an invariant interaction reasoning module, achieving state-of-the-art performance across various tasks. EqMotion is lightweight and effective for diverse motion prediction scenarios.
>
> MemoNet[2] is a trajectory prediction framework inspired by retrospective memory in neuropsychology. It utilizes memory banks to store representative past-future pairs and a trainable addresser to recall relevant instances, enabling more accurate and interpretable predictions. MemoNet achieves state-of-the-art results on multiple datasets, significantly improving prediction accuracy and diversity.
>
> FlowChain[3] is a normalizing flow-based model designed for fast and accurate trajectory prediction and density estimation. By leveraging conditional continuously-indexed flows (CIFs), it can achieve promising performance.
>
> We will include additional references on trajectory prediction in the related work section in our final version.
>
>
> [1] EqMotion: Equivariant Multi-Agent Motion Prediction with Invariant Interaction Reasoning. CVPR 2023.
> [2] Remember Intentions: Retrospective-Memory-based Trajectory Prediction. CVPR 2022.
> [3] Fast Inference and Update of Probabilistic Density Estimation on Trajectory Prediction. ICCV 2023.
>
>
>
> ***We sincerely appreciate your valuable comments and insightful suggestions. We hope our response effectively addresses your concerns.***

---

### Meta-Review · Area_Chair_7sW3 · 2024-12-20

**Metareview:**

The authors designed a novel approach for handling multiple tasks, which include trajectory prediction, imputation, and spatial-temporal recovery, for multi-agent motion analysis. All the four reviewers pointed out that the method is good and well designed, and thus all recommended acceptance. In the camera ready version, authors need to carefully improve the paper following reviewers' comments.

**Additional Comments On Reviewer Discussion:**

During the rebuttal, authors provided additional experiments, and additional details and justifications. Reviewers are generally satisfied with the reply.

---

### Decision · Program_Chairs · 2025-01-22

Accept (Poster)